



# Interaction between Atlantic cyclones and Eurasian atmospheric blocking drives warm extremes in the high Arctic

Sonja Murto[1], Rodrigo Caballero[1], Gunilla Svensson[1], and Lukas Papritz[2]

[1]Department of Meteorology and Bolin Centre for Climate Research, Stockholm University, Stockholm, Sweden
[2]Institute for Atmospheric and Climate Science, ETH Zürich, Zürich, Switzerland

**Correspondence:** Sonja Murto (sonja.murto@misu.su.se)

**Abstract.** Atmospheric blocking can influence Arctic weather by diverting the mean westerly flow polewards, bringing warm, moist air to high latitudes. Recent studies have shown that diabatic heating processes in the ascending warm conveyor belt branch of extratropical cyclones are relevant to blocking dynamics. This leads to the question of the extent to which diabatic heating associated with midlatitude cyclones may influence high-latitude blocking and drive Arctic warm events. In this study we investigate the dynamics behind 50 extreme warm events of wintertime high Arctic temperature anomalies. Classifying the warm events based on blocking occurrence within three selected sectors, we find that 30 of these events are associated with a block over the Urals, featuring negative upper-level PV anomalies over central Siberia north of the Ural Mountains. Lagrangian back-trajectory calculations show that almost 60 % of the air parcels making up these negative PV anomalies experience lifting and diabatic heating (median 11 K) in the six days prior to the block. Further, almost 70 % of the heated trajectories undergo maximum heating in a compact region of the midlatitude North Atlantic, temporally taking place between six and one days before arriving in the blocking region. We also find anomalously high cyclone activity (on average five cyclones within this five-day heating window) within a sector northwest of the main heating domain. In addition, 10 of the 50 warm events are associated with blocking over Scandinavia; the contribution of diabatic heating to these blocks is again around 60 % for six-day back-trajectories, of which 60 % undergo maximum heating over the North Atlantic but generally closer to the time of arrival in the block and further upstream relative to heated trajectories associated with Ural blocking. This study highlights the role of diabatic heating in high-latitude blocking dynamics and the importance of the interaction between midlatitude cyclones and Eurasian blocking as driver for Arctic warm extremes.

## 1 Introduction

The positive trend observed in global-mean surface temperatures is unequally distributed, with greater and more rapid surface warming seen over the Northern Hemisphere high latitude regions. This phenomenon, specifically observed in winter during recent decades, is known as Arctic amplification (e.g., Serreze and Barry, 2011; Cohen et al., 2014). As a result, dramatic





changes have been seen, such as Arctic sea-ice loss and decline in snow cover and continental ice sheets (e.g., Richter-Menge and Co-Authors, 2020; Simmonds, 2015).

The mechanisms responsible for Arctic amplification are still under discussion. A wide range of local and remote processes have been identified. Local processes comprise snow- and ice-albedo feedbacks (e.g., Screen and Simmonds, 2010a; Pithan and Mauritsen, 2014), enhanced ocean-atmosphere heat exchanges (e.g., Screen and Simmonds, 2010b; Boisvert et al., 2016), temperature feedbacks and related changes of water vapor content and cloud cover (e.g., Serreze et al., 2012), as well as circulation changes within the Arctic (e.g., Sorteberg and Walsh, 2008; Ding et al., 2017), while remote processes include

increased oceanic (Årthun et al., 2016) and atmospheric poleward transport of heat and moisture (e.g., Tjernström et al., 2015; Woods and Caballero, 2016; Naakka et al., 2019) mediated by anomalous large-scale circulation patterns outside the Arctic (e.g., Graversen, 2006; Woods et al., 2013; Liu and Barnes, 2015; Baggett et al., 2016). Especially the northward moisture transport from lower latitudes has emerged as a significant modulator of the surface energy balance in the Arctic, where intrusions of warm and humid air masses replace the cold and clear state by a warm and opaque state (e.g., Francis and Hunter,

2006; Doyle et al., 2011; Woods and Caballero, 2016; Mortin et al., 2016; Pithan et al., 2018).

Many of the aforementioned processes are highly episodic. For example, increasing northward moisture transport is the result of an increasing number of a few but intense moisture transport events, mainly occurring over the Pacific and Atlantic (Woods et al., 2013; Liu and Barnes, 2015; Naakka et al., 2019). Likewise, short-lived warm extremes have a disproportionate contribution to sea ice variability and loss (Boisvert et al., 2016; Cullather et al., 2016; Woods and Caballero, 2016; Moore,

2016; Binder et al., 2017; Kim et al., 2017; Wernli and Papritz, 2018). For this reason we focus here on episodic wintertime warm extremes.

## 1.1    Weather systems and importance of blocking

Northward moisture transport and Arctic warm extremes are favoured by specific circulation patterns. The large-scale flow associated with intense wintertime moisture transport events and warm surface temperature extremes in the high Arctic is

characterised by a poleward shift of the jet and a dipole in geopotential height over the North Atlantic, with a high over northern Eurasia and a low either in the high Arctic or along Greenland's east coast (Messori et al., 2018; Papritz, 2020; Fearon et al., 2020). This configuration is conducive to the transport of midlatitude air masses poleward across the Nordic Seas and deep into the Arctic basin. Such events are further favoured by the absence of widespread high pressure ridge over the Arctic ocean (Nygård et al., 2019).

The positive and negative geopotential height anomalies are the result of collocated anomalies in the frequency and pathways of particular weather systems. This includes the frequent northward displacement of the tracks of extratropical cyclones towards Greenland's east coast and extending into (or locally created in) the high Arctic (Sorteberg and Walsh, 2008; Messori et al., 2018; Fearon et al., 2020), anticyclonic Rossby wave breaking over the North Atlantic (Liu and Barnes, 2015), and blocking over Scandinavia or the Barents Sea and the Urals (Woods et al., 2013; Papritz, 2020). These circulation anomalies are often

concomitant, and a strong interplay between the cyclones and blocks has been established, where, e.g., the path of northward travelling cyclones is steered by blocking (Madonna et al., 2020; Papritz and Dunn-Sigouin, 2020).





Recent studies have highlighted blocking as a key driver of surface temperature extremes both in midlatitudes (Pfahl et al., 2014; Trigo et al., 2004; Zschenderlein et al., 2019) and in the Arctic (Tyrlis et al., 2019; Papritz, 2020). In particular, Ural blocking has been associated with Arctic sea ice loss as well as the so-called warm-Arctic-cold-Eurasian (WACE) temperature

pattern (Luo et al., 2016b, a; Tyrlis et al., 2019). Blocks can cause surface warming via adiabatic compression due to subsidence (Binder et al., 2017; Ding et al., 2017), the persistent transport of warm and humid air masses along its periphery (Woods et al., 2013; Papritz and Dunn-Sigouin, 2020), and in summer also enhanced incident shortwave radiation (Wernli and Papritz, 2018).

## 1.2  Blocking formation mechanisms

Given the importance of blocking for Arctic warm events, there has been a growing interest in examining the mechanisms

behind the formation and maintenance of high-latitude blocks, not least because of large biases in the representation of blocking in climate models (e.g., Tibaldi and Molteni, 1990; Woollings et al., 2018). The dynamical mechanisms driving the formation and maintenance of blocks are generally not fully understood and as a consequence, there is no agreement on a unique definition of blocking. Thus, a range of different blocking identification methods exists, each one emphasizing different characteristics of blocking such as flow reversal or upper-level potential vorticity (PV) anomalies (e.g., Tibaldi and Molteni, 1990; Pelly and

Hoskins, 2003; Schwierz et al., 2004; Croci-Maspoli et al., 2007; Tyrlis and Hoskins, 2008).

Classical, dry-dynamical theories for blocking include global theories emphasising the influence of planetary-scale dynamics including Rossby wave trains and their interaction with the background flow and topography (e.g., Grose and Hoskins, 1979; Hoskins and Karoly, 1981; Reinhold and Pierrehumbert, 1982), as well as local theories pointing out the importance of Rossby wave breaking (Masato et al., 2012) and slow-moving synoptic-scale eddies located upstream of blocks (Colucci,

1985) in blocking dynamics. Coupling of low- and high-frequency forcing and their relative importance are also highlighted by Nakamura et al. (1997).

In general, blocking can be characterized as an extended ridge, represented by high potential temperature ($\theta$) anomalies on the dynamical tropopause (PV isosurface) or low PV anomalies on an isentrope. This latter characterization of blocking is particularly useful as it relates blocking to conserved quantities (PV, $\theta$) in adiabatic flow (Hoskins et al., 1985; Woollings et al.,

2018). Despite the quasi-stationary nature of the blocking patterns, often the upper-level flow in blocks is highly dynamic. For example, low-PV air in the upper troposphere is constantly refuelled via the isentropic advection of air from regions with climatologically lower PV values (Hoskins et al., 1985).

In addition to the dry dynamical processes described above, recent studies have emphasized the importance of moist processes associated with the cross-isentropic transport of low-PV air into the blocking region. More precisely, cloud-diabatic

processes in the strongly ascending warm conveyor belt of extratropical cyclones have been found to contribute substantially to the formation and maintenance of the negative upper-level PV anomaly by transporting low PV air cross-isentropically from the lower troposphere into the upper troposphere (e.g., Croci-Maspoli and Davies, 2009; Madonna et al., 2014, 2015; Pfahl et al., 2015; Steinfeld and Pfahl, 2019).

Our goal in this study is to investigate the dynamics behind Arctic warm extremes. Specifically, we aim to address the

following questions:





- What is the role of blocking in driving these extreme warm events?

- Are there regional differences in the circulation patterns among the different warm extremes?

- How important are diabatic processes in driving the blocks?

- How do cyclones and blocks interact during these warm events?

As discussed in Sect. 2, which provides an overview of the data and methods used in this study, we focus on the top 50 high Arctic warm extreme events previously examined in Messori et al. (2018). Our results are presented in four sections (Sects. 3-6), beginning with an illustrative example of a sequence of three temporally close warm extreme events and their dynamical drivers (Sect. 3). Sect. 4 investigates the role of blocking for all of the 50 warm extremes, presented in their corresponding event clusters. Subsequently, the dynamics of the blocking events are discussed in Sect. 5, including an examination of blocking

trajectory characteristics. This is followed in Sect. 6 by a discussion of the interaction between midlatitude cyclones and the blocks associated with warm events. We finalize our study with a discussion in Sect. 7, and summarize our key findings in the concluding Sect. 8.

## 2   Data and Methodology

The analysis in this study is based on 6-hourly ERA-Interim reanalysis data (Dee et al., 2011) provided by the European Centre

for Medium-Range Weather Forecasts (ECMWF). We focus on the Northern Hemisphere mid- and high- latitudes for the extended winter season, November through March (NDJFM). The original spatial resolution of ERA-Interim, approximately 80 km, vertically on 60 model levels, is used for most parameters, although for some (potential vorticity and geopotential height) we use data horizontally interpolated to a $1° \times 1°$ grid and on pressure levels in the vertical. Anomalies are defined as deviations from the calendar-monthly climatology averaged over the 38-year period used in this study (1979-2016) unless

otherwise mentioned.

### 2.1   High Arctic warm extreme events

We focus on the same 50 extreme warm events of wintertime high Arctic 2-meter temperature (T2m) anomalies previously identified in Messori et al. (2018). We briefly summarize the identification method here for the reader's convenience. Warm anomalies are computed w.r.t. a daily climatology from which the non-linear warming trend seen in the Arctic (Cohen et al.,

2014) is removed by applying a 21-day running window on a 9-year running-mean calendar-day climatology. Mean Arctic temperature anomalies are then obtained by area-weighting and averaging the daily values over the polar cap ($> 80°$ N). Finally, aiming at retaining persistent deviations from the climatology and avoiding double counting of events, a further 5-day running-mean filter is implemented and the most anomalous days with at least one week time difference are chosen. The top 50 warm extremes are then selected, ranked by decreasing temperature anomaly. In the rest of the paper, individual events are

identified by their rank (e.g. event 3 is the third-warmest event).

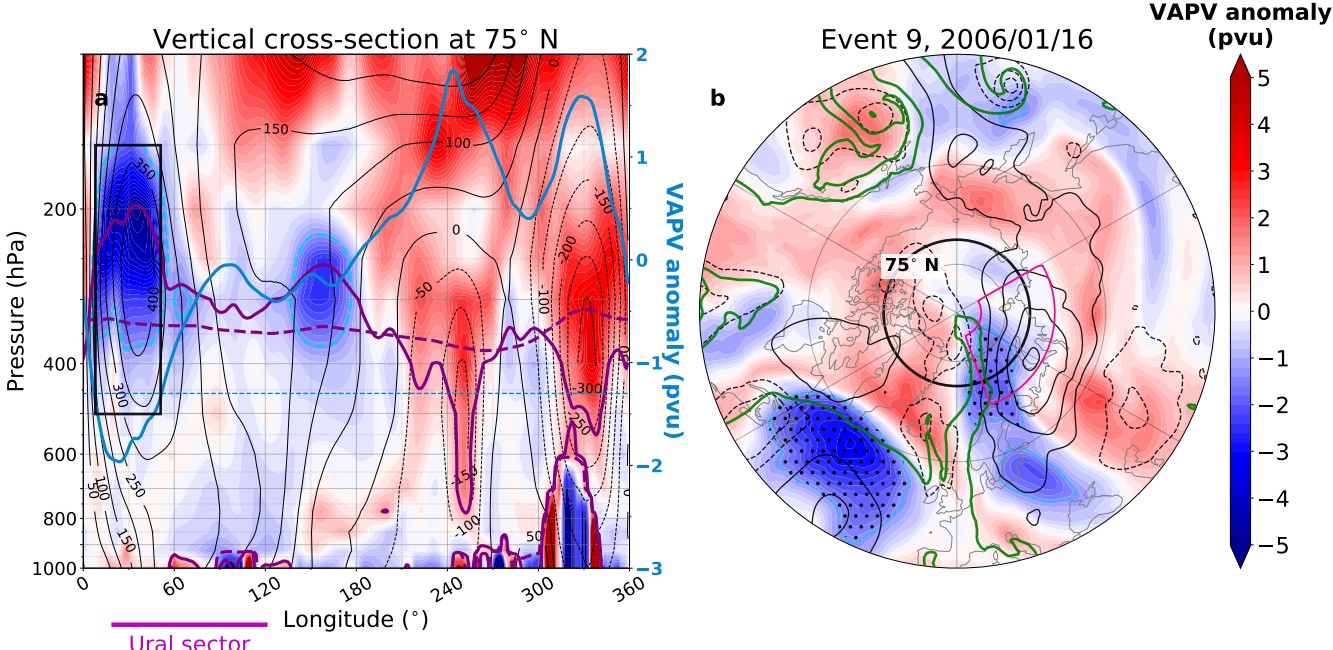

**Figure 1.** An illustrative example of the blocking identification algorithm for 12 UTC 16 January 2006, one day prior to event 9. (a) Vertical cross-section along 75° N showing PV anomaly (shading, -1.3 pvu isoline highlighted by cyan, dashed line), tropopause at 2 pvu (purple solid), climatological tropopause (purple dashed), and geopotential height anomalies (black contours every 50 gpm). Also shown is the VAPV anomaly and the threshold of -1.3 pvu (light blue solid and dashed contour, respectively, scale on the right). The Ural sector is indicated by a magenta horizontal line below the figure and the black box highlights the identified Ural block. Note the logarithmic scale in the vertical. (b) VAPV anomaly (shading) overlaid with SLP anomalies (hPa, black contours every 10 hPa, negative values dashed, zero contour not shown), total column water ($5\,\mathrm{kg\,m^{-2}}$ isoline, green solid) and VAPV anomaly (-1.3 pvu, cyan dashed). Dotted areas show the identified blocking regions and the black line and the magenta box indicate the location of the cross-section in (a) and the Ural sector, respectively.

It is worth mentioning that the four most extreme events (January 2006, March 1992, November 2016 and January 2000) are well distinguished from the other extremes, obtaining T2m anomalies greater than 10 K. A detailed overview of the 50 warm extreme events is presented in Sect. 4.

## 2.2 Blocking identification

Quasi-stationary atmospheric blocks are identified using a PV-index method, which is based on upper-level (150-500 hPa) negative vertically averaged PV anomalies, following the algorithm by Schwierz et al. (2004). The 6-hourly vertically averaged PV (hereafter VAPV) anomaly fields, computed with respect to the monthly climatology and temporally smoothed using a 2-day running-mean filter, are used as input data for the blocking index calculations. This dynamically-based method cap-



tures the core of the PV anomaly located in the upper troposphere, enabling a more comprehensive analysis of its dynamical
characteristics, lifetime, maintenance and evolution.

A temporal and spatial overlapping criterion is used to find each blocking life-cycle. Here, at each time step, blocking masks
are determined from the VAPV field where it falls below a threshold value of -1.3 pvu (1 pvu = potential vorticity unit =
$10^{-6}\,\mathrm{m^2\,s^{-1}\,K\,kg^{-1}}$), which is based on previous studies (e.g., Croci-Maspoli et al., 2007; Pfahl et al., 2015; Steinfeld and
Pfahl, 2019; Lenggenhager and Martius, 2020). Then, these masks are connected in time if at two consecutive time steps they
overlap by at least 70 %. We then identify a block if the temporally connected masks persist for at least 5 days. Applying the
blocking algorithm for the whole study time period results in fields indicating the presence of a block at each grid point and
6-hourly timestep and its unique identification number .

Additionally, the life stage of each individual blocking life cycle is quantified by the D-index. This index attains values
between 0 and 1, and is defined as follows:

$$\textbf{D-index} = \frac{\text{Time since blocking onset}}{\text{Total blocking duration}}. \tag{1}$$

Figure 1 illustrates the blocking identification method applied to event 9, where the black box in Fig. 1a highlights the iden-
tified upper-level blocking region in which the above-mentioned criteria are fulfilled. Within this box, the elevated tropopause
coincides with positive and negative geopotential height and PV anomalies, respectively. The block north of the Ural mountains
in Fig. 1b has a northwestward tilt relative to the region of positive sea-level pressure (SLP) anomalies observed at the surface,
and it is accompanied by a northward flow of moist air to the west of the block.

## 2.3   Trajectory calculation

To further quantify the importance of different processes leading up to the identified blocks, we compute 10-day three-
dimensional kinematic back-trajectories using the Lagrangian Analysis Tool (LAGRANTO; Sprenger and Wernli, 2015). LA-
GRANTO calculates trajectories $\mathbf{x}$(t) using three-dimensional wind fields on reanalysis model levels and a 30 minute computa-
tional time step. Trajectory starting positions are selected based on the blocking mask, re-gridded to match the same horizontal
resolution as the reanalysis data. More specifically, trajectories are initialized at every grid point within the blocking mask,
equally at every $80\,\mathrm{km} \times 80\,\mathrm{km}$ grid, every $50\,\mathrm{hPa}$ from 500 to $150\,\mathrm{hPa}$. Starting times are daily at 12 UTC. For the analysis,
however, blocking preceding a warm event is being represented by one specified day within the narrow $-3$ to $-1$ day window
before each warm event, as will be explained in Sect. 4. Only starting points with $< 1\,\mathrm{pvu}$ are used in the interest of studying
tropospheric air. Additionally, various quantities are traced along each trajectory, such as PV, $\theta$, PV anomaly (here calculated
with respect to the 10-d running-mean climatology) and specific humidity ($Q$).



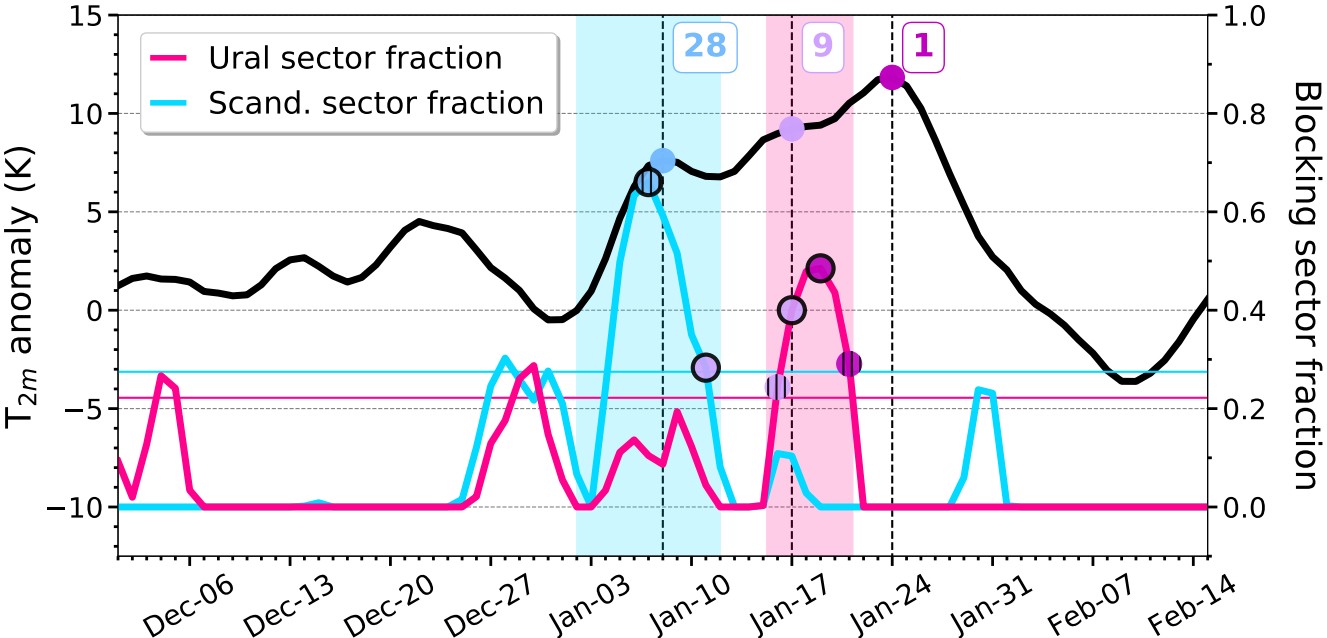

**Figure 2.** Daily variation of high-Arctic area averaged T2m anomalies (black), blocking fraction (based on the area fraction of the sector that intersects with a block as identified by the algorithm described in Sect. 2) area-averaged over the Ural (magenta) and Scandinavian (cyan) sectors during 1 Dec - 15 Feb 2006. Three Arctic warm events are marked with vertical black dashed lines with event-ID indicated in boxes to the right of each event-line. The magenta and cyan horizontal thin lines mark the $90^{th}$ percentile thresholds of the daily blocking fractions and the lifetime of two blocks are indicated by magenta and cyan shading for Ural and Scandinavian blocks, respectively. The timings of the maximum blocking fraction within three (corresponding to the time of trajectory initialization) and six days prior to each event are shown by hatched and solid circles, respectively, where light blue circles refer to event 28, light purple markers to event 9 and magenta markers to event 1. The x-ticks are shown at 1200 UTC.

## 2.4 Cyclone tracking method

Cyclone tracks are calculated using the algorithm developed by Murray and Simmonds (1991). Tracking is done on mean sea-level pressure data (MSLP) north of 30° N. Same instruction parameters as listed in Table 1 in the study by Uotila et al. 160 (2009) are selected. Only cyclones lasting for at least 24h are included in this study.

## 3 Case study of three successive Arctic warm extreme events in January 2006

Before proceeding to the climatological analysis of the dynamics behind the extreme warm events, a case study for winter 2005/2006 is presented to illustrate the interaction between Atlantic cyclones and Eurasian blocking in the lead-up to Arctic warm extremes. The winter 2005/06 in Europe was dominated by blocking regimes, especially in January 2006 with cold





anomalies observed over continental Europe and Russia (Croci-Maspoli and Davies, 2009), which were accompanied by periods of heavy snowfall over central Europe (Pinto et al., 2007). Conversely, positive DJF T2m anomalies occurred over the North Atlantic, North America, and the polar regions (Croci-Maspoli and Davies, 2009).

     The algorithm of Messori et al. (2018) identifies three high Arctic warm extreme events during January 2006: events 28, 9 and 1, occurring on the 8th, 17th and 24th and attaining a high-Arctic averaged T2m anomaly of 8, 9 and 12 K respectively

(see Fig. 2). It is clear that though they are classified as individual events, the three peaks are actually local maxima within a constantly increasing temperature anomaly until reaching the most extreme event. The analysis below suggests that they could also be thought of as sub-events in a single, persistent warm extreme supported by two separate blocking events that manifest in two different sectors, one near the Ural mountains and another over Scandinavia; these sectors are further discussed in Sect. 4.

### 3.1 Synoptic situation

On 3 January (Fig. 3a), a ridge (high $\theta_{2pvu}$ values) extends over the North Atlantic and further eastward. An upper-level block is identified northwest of the United Kingdom (dotted shaded region). Two days later (Fig. 3d), the upper-level block has moved northeastwards and now extends over Scandinavia. Meridional flow in the North Atlantic prevails, bringing warm and moist air northwards driven by the dipole in the SLP field. Two days later (lag = −1 day, Fig. 3g), the block maintains its position over Scandinavia and the moist and warm air continues to penetrate deeper into the Arctic. The peak in blocking fraction in

the Scandinavian sector (Fig. 2) occurs at the same time. At the peak of the warm event (Fig. 3j), the block is still located over Scandinavia and steers the flow northwards, but is then slowly decaying (see Fig. 2) and moving southwards (not shown).

     Three days after the first peak, on 12 January, a new event begins (Fig. 3b). The Scandinavian block is replaced by a low-pressure system, the remaining block (dotted region over Poland) directs the flow around the Urals and enters into the Arctic over the Barents and Kara Seas. However, two days later (Fig. 3e) the high over the Urals is strengthened and a low-pressure

center is formed east of Greenland. At this time there is no substantial blocking identified (Fig. 2). The meridional flow strengthens as a consequence of the stronger dipole over the North Atlantic, favouring deep penetration of warm and moist air into the high Arctic at lag −1 day (Fig. 3h). Note that a new block is identified over the Arctic Ocean northeast of Scandinavia (dotted area, the Ural sector is shown in magenta). The upper-level block tilts northwestward from the surface high pressure over the Urals and northern Siberia (see also Fig. 1 and discussion in Sect. 2.2). The block expands eastwards, covering large

parts of the Ural sector at the day of the warm event (Fig. 3k and Fig. 2) and the northwestward tilt remains. A strong positive response in the temperature anomaly is also seen to the north/northwest of the block and a negative one to the south/southeast of the block (Fig. 3n), resembling the WACE pattern.

     The next event occurs one week after the second temperature anomaly peak, also influenced by the same established Ural block. The peak in the Ural sector fraction reaches its maximum value five days prior to the warmest event (see Fig. 2), after

which the Ural sector fraction decreases such that three days prior to event 1 a Eurasian block is no longer detected (Fig. 3f). However, the northward advection of warm and moist air continues with the main pathway east of Greenland (Fig. 3c, f, i). After this period of increased blocking activity over Eurasia, the temperature anomaly decreases (see Fig. 2) as the flow becomes more zonal (not shown).




The spatial distribution of the T2m anomaly at the time of the extremes (Fig. 3m-o) show a dipole between anomalously cold
Eurasia and warm Atlantic and Arctic and the negative anomalies coincide with contours of higher SLP, further supporting the
findings in Croci-Maspoli and Davies (2009).

## 3.2 Linkage between Atlantic cyclones and blocks over Eurasia associated with the three warm events in January 2006

To explore whether there is a link between the identified blocks and cyclones, we present three sets of back-trajectories (Fig. 4)
initialized from the two identified Eurasian blocks: the Scandinavian block one day prior to event 28 (first column), and from
the Ural block influencing both of the last two events; one day prior to event 9 (second column) and three days before event
1 (last column), times identical with the synoptic situation presented previously in Fig. 3 (g), (h) and (f), respectively. At
the time of trajectory initialization for all three events (see Fig. 4a-c), a narrow band of moist air is intruding into the Arctic
west of the block. Also, we consider cyclones and the locations of the trajectories at the time of maximum heating. Note that
only trajectories that experience diabatic heating (substantial increase in $\theta$) are shown, constituting 52 %, 73 % and 5 % of all
computed trajectories for each of the three events, respectively. A more detailed definition of the trajectory classification is
presented later in Sect. 5.1.

The majority of the back-trajectories started from the Scandinavian block (Fig. 4a), 2.5 weeks prior to the most extreme
event, reside at lower altitudes south of 50° N over eastern North America and in the North Atlantic six days earlier (Fig. 4g).
Only 2 % of all heated trajectories experience their maximum heating (green density contours) already at this stage, mainly in
the vicinity of cyclones over the eastern Pacific and southwestern North Atlantic. The air parcels experience maximum heating
at various times during their journey into the blocking region, though with a peak (13 % of all heated trajectories) over the
North Atlantic 3.5 days prior to arrival in the block (Fig. 4d). A few of the back-trajectories that pass over the Mediterranean
get heated and lifted one day later (not shown). After the maximum, all blocking trajectories reside in the upper troposphere
for several days before reaching the block (Fig. 4a).

The remaining panels show back-trajectories from the same short-lived (persisting for only six days) Ural block, initialized
at different life-stages of the block with respect to the peak of event 9 and 1. As for the event 28, most of the heated trajectories
initialized at the onset stage of the Ural block (Fig. 4b), eight days prior to event 1, reside around or south of 40° N in the
northern part of the subtropical high over the ocean six-days prior to arrival into the block (Fig. 4h). Also here, a small fraction
of the computed trajectories experience their maximum heating at this early stage, mainly over the eastern Pacific. However,
the majority of the maximum heating occurs within the North Atlantic at different times from four days before arrival, with
a peak (15 % of all heated trajectories) observed 2.5 days prior to arrival into the blocking region (Fig. 4e). The trajectories
follow a fairly coherent path and move quickly once they reach the upper troposphere. Again, the lifting occurs primarily in
the vicinity of a mid-latitude cyclone, located northwest of the center of maximum heating.
In contrast, capturing only 5 % of heated trajectories initialized from the decaying stage of this Ural block three days prior
to event 1 (Fig. 4c), the time window where the maximum heating within the North Atlantic occurs is narrowed to the first two
days from the end points of the trajectories. Already 15 % of all trajectories experience heating over the North Atlantic six-days





prior to arrival (Fig. 4i) and 21 % a few days later (maximum peak at lag −4.25 day, Fig. 4f), after which all trajectories again reside in the upper-troposphere for several days before arriving in the block. Again, we observe a cyclone located to the north

of the region of maximum heating (Fig. 4f, i), which in fact is the same cyclone contributing to the late lifting in Fig. 4e, though at different times and spatial locations. This suggests that a substantial portion of the lifting for these Ural blocking trajectories is accomplished by just one cyclone in the North Atlantic. The observed negative correlation between blocking life-stage and the timing of maximum heating over the North Atlantic—i.e. younger blocks experience maximum heating at later times— will be discussed more generally in Sect. 5.

In summary, we showed in this case study that two significant blocks, one over Scandinavia and one over the Urals, contributed to three close-in-time warm extreme events in January 2006, the latter of which is the most extreme in the entire observational record. Additionally, a large fraction of six-day back-trajectories traced from the blocks experienced heating, in agreement with Pfahl et al. (2015) and Steinfeld and Pfahl (2019). We also saw that the majority of the heating and lifting is accomplished by a small number of cyclones in the North Atlantic.

## 245   4  Classification of warm events by type of blocking

Our case study (Sect. 3) highlights the importance of Ural and Scandinavian blocking for some of the most extreme Arctic warm events in the observational record. This raises the question of whether blocking plays a similar role in all the top-50 Arctic warm extreme events introduced in Sect. 2.1. We address that question in this section. We start by examining the 50 events collectively and assess whether or not they were preceded by significant blocking, and then proceed to classify the events

by the type of blocking observed. Additionally, we discuss events that are temporally close to each other and their relation to blocks.

    Figure 5a shows composite VAPV anomalies for all 50 events, averaged over the 3 days preceding the peak of the events. The composite shows a prominent negative anomaly in the Ural sector and a weaker negative anomaly over Scandinavia, suggesting that blocking in these regions is indeed a common feature to many of the events. The corresponding SLP anomalies

(black contours) show a pattern consistent with that shown in Messori et al. (2018), with a high over Siberia and the Barents and Kara seas and a low over Greenland, conducive to advection of warm, moist air into the high Arctic, as indicated by values of total column water vapour typical for mid-latitudes penetrating into the Arctic.

    Detailed inspection of VAPV maps for each individual event shows considerable variability among events, however: some events feature isolated VAPV anomalies over the Urals, some over Scandinavia, and some an extended anomaly covering both

regions; other events show generally weak anomalies everywhere, and a small number of events show very strong negative anomalies in the Pacific sector near Bering Strait.

    To translate this subjective inspection into an objective classification of blocking types, we select 3 sectors capturing the locations where prominent negative VAPV anomalies are most frequently found: Ural (20 - 120° E, 70 - 85° N), Scandinavia (0 - 50° E, 55 - 70° N) and Pacific (180 - 120° W, 55 - 85° N) sector (Fig. 5a, Table 1). We then define a blocking index for each

sector based on the area fraction of the sector that intersects with a block as identified by the algorithm described in Sect. 2.2.





Specifically, at each 6-hourly time step we compute the total area of the grid cells within the sector belonging to a block, divide by the total sector area and take a daily mean to yield a daily sector blocking index. We then relate a warm event to a blocking in a specific sector if the corresponding sector index exceeds its 90$^{\text{th}}$ percentile value at some point over the 6 days prior to the warm event. For the Pacific sector, we use the 99$^{\text{th}}$ percentile in order to capture only the strongest blocking in this frequently

blocked area.

This procedure partitions the 50 warm events into 5 distinct clusters (Fig. 5b-f and Fig. 6): 3 "pure" clusters (Ural, Scandinavian and Pacific, containing 18, 7 and 2 events, respectively), one mixed Ural/Scandinavian cluster (12 events), and a residual cluster of 11 other events not associated with blocking in any of the sectors. The partitioning between the Ural, Scandinavian and mixed clusters is illustrated in Fig. 6b. Note that one of the residual events (coral markers) achieves a Ural blocking index

marginally above the threshold, but is not classified as a Ural event because detailed examination of the VAPV anomalies for that event shows that blocking extends over the entire polar cap and Greenland, making it qualitatively different from the other 18 Ural events.

Circulation patterns composited over the various clusters are shown in Fig. 5b-f. Compositing over the 18 pure Ural events (Fig. 5c), or over the 30 events with Ural blocking (i.e. combining the Ural and mixed Ural/Scandinavian clusters, Fig. 5b)

shows patterns very similar to that for composites over all 50 events, though with greater amplitude. In these cases, the maximum negative VAPV anomaly is concentrated in and around the Ural sector, while the SLP anomaly shows a clear dipole straddling the pole and promoting warm advection from the North Atlantic into the high Arctic. The Scandinavian composite (Fig. 5d) has peak VAPV anomaly over Scandinavia, as expected, while the SLP again shows a dipole structure with a high over Scandinavia, partly extending over to northern Siberia, and a low over Greenland, again promoting advection from the

North Atlantic. The Pacific composite shows very strong negative VAPV anomalies over Alaska, and the SLP anomalies show a dipole with a high over Alaska and a low over eastern Siberia. This pattern supports warm advection from the North Pacific, reflected in the structure of the total column water field (green contour) showing intrusion of mid-latitude air into the Arctic from the Pacific. Note that the 30 % blocking frequency contour (light blue) nicely encloses regions of peak negative VAPV anomaly in each of the aforementioned cluster composites. Finally, the 11 residual events show weak VAPV and SLP anoma-

lies with a structure similar to that in the pure Ural composite. This resemblance suggests that many of the residual events are in fact weak Ural events, and might have been classified as such had we used a lower VAPV threshold in the blocking identification algorithm. However, in the remainder of the paper we focus on the higher-amplitude events in the other clusters.

As noted in Sect. 3, our case study shows that two successive warm extremes occurring in rapid succession can in fact be driven by a single blocking event. We find that this occurs 5 times within our top 50 warm extremes. These 5 pairs of warm

events (connected by black lines in Fig. 6a) all occur within 13 days of each other, and the second event is always warmer than the first. The criteria used to select the warm extremes (following Messori et al., 2018) arbitrarily stipulate that two consecutive events are considered independent if they are separated by more than 1 week. As also noted in Sect. 3, the more physically-based blocking perspective taken here suggests that these pairs of events could also be thought of as single, long-lasting warm events driven by a single blocking event.



## 5 Dynamics of blocking events

The previous section showed a strong association between blocks and extreme Arctic warm events. Here, we take a closer look at the dynamics behind these blocking events, focusing on the sources of low-PV air as identified by Lagrangian back-trajectories initialised within the blocks (Sect. 2.3). We are particularly interested in the contribution of diabatic heating within mid-latitude cyclones (Pfahl et al., 2015; Binder et al., 2017).

### 5.1 Ural blocking

We focus first on blocking in the Ural sector, combining the 18 pure Ural blocks with the 12 mixed Ural/Scandinavian events (Fig. 5b). For the latter, the blocking algorithm identifies two disjoint blocking regions—one over the Urals and one over Scandinavia—in 6 cases; in these cases, trajectories are initialised only from the Ural block. In the remaining 6 cases, a single block extending over both sectors is identified. We apply an additional geographical mask in order to initialize trajectories only over the portion of the block residing over the Ural mountains (mainly north of 60° N). Furthermore, a few of the pure Ural events also obtain scattered blocking regions, for which we apply the same method as described above. The exact masks used for a total of seven Ural events are shown by the footnotes after each event number in Table S1 in the supplemental material (text section S1). Back-trajectories are started at the time of maximum Ural blocking fraction within the interval 3 to 1 days before the peak of each warm event. We compute a total of 97405 trajectories initialized three, two and one day prior to each warm event for 12, 6, and 12 of the 30 Ural events, respectively.

Following previous work (Pfahl et al., 2015; Steinfeld and Pfahl, 2019), we assess the role of diabatic heating by examining the change in dry potential temperature ($\Delta\theta$) along trajectories as follows. We identify the absolute maximum $\theta$ along the trajectory and find the (positive) difference $\Delta\theta_+$ between this maximum and the previous minimum $\theta$. Similarly, we find the (negative) difference $\Delta\theta_-$ between the absolute minimum and the subsequent maximum. If $\Delta\theta_+ > 2$ K, the trajectory is classed as "heated" and $\Delta\theta = \Delta\theta_+$; otherwise, the trajectory is classed as "not heated" and $\Delta\theta$ is set equal to either $\Delta\theta_+$ or $\Delta\theta_-$, whichever has greater absolute value.

Figure 7a shows histograms of $\Delta\theta$ for all 30 events using backward-trajectories of increasing length; the inset panel shows the fraction of heated trajectories as a function of trajectory length. The histograms are bimodal by construction, with separate positive and negative lobes. As trajectory length increases the two modes move further apart: air parcels experience greater heating and cooling the longer they are traced. The fraction of trajectories in the heated class rises rapidly from less than 10 % at 1 day to almost 60 % (59 %, 57323 trajectories) at 6 days but changes little thereafter, implying that 6 days is a reasonable trajectory length when considering diabatic heating effects in Ural blocking.

Figure 7c shows the time evolution of $\theta$ and pressure for trajectories in the 6-day heated regime. At 6 days before arrival, air parcels are almost entirely below the 500 hPa level (warm colours), with the median trajectory (thick black line) close to 700 hPa and $\theta$ around 300 K, typical values for mid-latitude air (Hoskins et al., 1985). In the subsequent days they warm and rise into the upper troposphere. The median trajectory rises by around 300 hPa in 3 days, but the ascent rate for individual trajectories is considerably more rapid: the maximum 2-day pressure drop for these trajectories has a median value around



**Table 1.** Sectors and regions used in the study for the event clustering (right) and cyclone frequencies and the heating domain (right).

| Sector | coordinates | Region | coordinates |
|---|---|---|---|
| Ural | $(20\text{-}120)^\circ$ E, $(70\text{-}85)^\circ$ N | North Atlantic heating domain (A) | $(80\text{-}0)^\circ$ W, - |
| Scandinavia | $(0\text{-}50)^\circ$ E, $(55\text{-}70)^\circ$ N | North-West Atlantic cyclone sector (C) | $(60\text{-}10)^\circ$ W, $(50\text{-}70)^\circ$ N |
| Pacific | $(180\text{-}120)^\circ$ W, $(55\text{-}85)^\circ$ N | high Arctic | $\geq 80^\circ$ N |

400 hPa (Fig. 8e). Further, the median humidity change along these trajectories is a drying of around $4\,\mathrm{g\,kg^{-1}}$ (Fig. 8i); condensation of this amount of water vapour yields an isobaric warming of around 10 K, comparable to the median heating

of about 11 K for 6-day trajectories shown in Fig. 8a. Overall, these results are consistent with the view that diabatic heating occurs principally through latent heat release in the rising branches of cyclones (Madonna et al., 2014).

Trajectories in the no-heating regime, on the other hand, are almost entirely in the upper troposphere on day $-6$ (Fig. 7e); they cool diabatically at a median rate of around $1.3\,\mathrm{K\,day^{-1}}$ (Fig. 8a), consistent with typical radiative cooling rates in the upper troposphere, and subside by around 140 hPa as they travel to the blocking region.

The results above show that diabatic heating plays a major role in the identified Ural blocking events, at least when averaging over all 30 cases. There is considerable variability among cases, but 23 of them (77 %) have a 6-day heated fraction in excess of 40 % (Fig. 9, red markers). Cases 1 and 8 both have a very low heating fraction, less than 10 %. Interestingly, these are both cases in which a single blocking event generates 2 consecutive warm events (see Fig. 6). They are thus examples of long-lived blocks, where low-PV air has been recirculating for several days after diabatic heating (Steinfeld and Pfahl, 2019). Low heating

fractions is also consistent with the blocks being in the decaying stage of their life-cycles (e.g., Pfahl et al., 2015; Steinfeld and Pfahl, 2019). Averaging over the per-event defined 6-day heating fractions, we obtain a slightly lower percentage of 54 %, mainly due to the influence of these two events.

We turn now to the question of where geographically the diabatic heating and ascent of air parcels feeding into Ural blocks takes place. For each trajectory in the 6-day heated class, we identify the location of peak diabatic heating rate as the point

of maximum $\theta$ increase over a six-hour period between 6 to 1 days before arrival at the blocking region. To filter high frequency noise, we first smooth the one-hourly $\theta$ values with a six-hourly running-mean. Fig. 10a presents the resulting spatial distribution of peak heating. The bulk of the trajectories (68 %) undergo peak heating in the Atlantic sector (dashed lines in the figure, see also Table 1, right), particularly in the mid-latitude central and eastern parts of the basin just west of the British Isles. This structure differs markedly from the climatological distribution presented in Steinfeld and Pfahl (2019, their Fig. 7),

which shows heating concentrated in the western Atlantic off the North American coast: by focusing on Ural blocking, we are selecting a relatively infrequent subset of trajectories shifted north-eastward toward the Ural sector. In addition, a substantial number of trajectories undergo heating over the eastern Mediterranean, and smaller numbers over the North American continent and the central Pacific. The relative fraction of trajectories experiencing maximum heating in the North Atlantic increases up to 86 % when applying the stricter criteria consistent with warm conveyor belts (WCBs), discussed in more detail in the





supplemental materials (Sect. S2). The distribution shown in Fig. 10a strongly points to a connection between Ural blocking
and diabatic heating in cyclones within the main North Atlantic storm track, as we will explore in Sect. 6 below.

Figure 11a shows the time evolution of trajectory density for trajectories undergoing maximum heating in the Atlantic sector
(note that only 29 of the 30 events are included here, since trajectories initialized from the block associated with event 5
originate from the Pacific and are advected over Siberia). Two days before peak heating, trajectories are concentrated in the
lower troposphere over the subtropical to mid-latitude western Atlantic. SLP composites at this time (shading in Fig. 11a)
show a pattern corresponding to the positive phase of the North Atlantic Oscillation (NAO). This pattern is consistent with the
findings of Messori et al. (2018), which show that Arctic warm extremes are preceded by a NAO+ and a northward shift of the
North Atlantic jet. The positive lobe of the SLP pattern advects warm, moist subtropical air which then travels north-eastward
over the central Atlantic and rises rapidly, with the bulk of the ascent accomplished over the 24-hour period straddling the time
of peak heating (Fig. 11b). Once in the upper troposphere, air parcels are advected over the subsequent days toward the Ural
sector. Similar behaviour regarding the pressure evolution is observed when restricting the magnitude of ascent for the heated
trajectories, though with larger pressure differences between the time of maximum heating and its near time steps (Fig. S1 in
the supplemental material).

We have now shown that the majority of the Ural blocking trajectories undergo maximum heating and lifting in the Atlantic
sector (Fig. 11), mostly in the period between 1 and 6 days prior to arrival into the blocking region (Fig. 7c). Frequency
distributions over the time lag of maximum heating in the Atlantic sector, performed for individual trajectories per event,
reveal that maximum heating is typically concentrated in one (70 %) or two (20 %) bursts, the former temporally taking place
either at early (longer than 3.5 days, 30 %) or late (shorter than 3.5 days, 17 %) lags or around lag −3.5 day (23 %). The
remaining events experience maximum heating almost evenly within the 5-day window or lack maximum heating over the
Atlantic (one event). Based on the frequency distributions, we then define for each event a time of peak heating, i.e. one lag
when majority of the heated trajectories experience maximum heating.

The maximum heating on average occurs at approximately 4 days prior to arrival in the blocking region (median lag of −3.5
days, Fig 12b). We refer the reader to the supplemental material for a discussion of the change in time of maximum heating
by applying additional pressure criteria (text section S2 and Fig. S2 in the supplemental material), where we show a shift of
maximum heating to later times, closer to the blocking region, when approaching the definition of WCBs.

We further find a correlation of −0.42 between the time of peak heating and the life stage of the block (D-index, Eq. (1),
visualized in Fig. 12a. This implies that trajectories initialized from older blocks (higher D-index) generally experience peak
heating in the Atlantic sector at early lags, many days prior to arrival at the block, and vice versa (as also seen for the Ural
blocking trajectories presented in the case study in Sect. 3). As the sector blocking fraction partly reflects the size of the block,
we observe that blocks at their mature stage (D-index values close to 0.5) usually obtain high sector fractions, as seen by the
darker coloring of the markers in Fig. 11a. This is consistent with the climatology of the evolution of blocking size presented
by Croci-Maspoli et al. (2007).

Lastly, when considering the time of peak heating for all heated trajectories per each Ural event, not only for those experi-
encing maximum heating within the Atlantic sector, we find a weaker correlation of −0.33 between the time of peak heating





and the D-index. This weaker dependency arises if we obtain different times of peak heating for an event for these two cases, especially when only a minority of the air parcels experience maximum heating in the Atlantic sector. The event-wise defined peak lags for both cases discussed above, as well as the relative fractions of heated trajectories being heated over the Atlantic and the D-index related to each block are listed in the supplemental material (Sect. S1) in Tables S1-S4.

### 5.2   Scandinavian blocking

This section examines the dynamics behind Scandinavian blocking events, following the same approach as for Ural events in the previous section. Back-trajectories are initialized from blocks observed over the Scandinavian sector (Table 1, see also light blue sector in Fig. 5d). In Fig. 6a, markers overlaid with a small black dot denote the ten events included here, namely six pure Scandinavian events and an additional four events from the mixed Ural/Scandinavian cluster which show an isolated blocking region over Scandinavia. To enable comparison with the Ural case and analyse blocking trajectories initialized within

lags $-3$ to $-1$ days relative to trajectory starting points, events where the block decays more than 3 days prior to the peak warm anomaly are excluded. One of the pure Scandinavian events (event 24) features two separate blocks, which are treated separately here. This leaves us with 10 Scandinavian events consisting of 11 blocks for the Lagrangian analysis performed in this section. One event obtains scattered blocking regions, to which a geographical mask is applied in order to retain only the region of the block located over Scandinavia (see footnote in Table S3 in the supplemental materials, Sect. S1). We compute

a total of 32255 trajectories initialized on the day of maximum Scandinavian blocking fraction: three days prior to the Arctic warm event in five cases, and one day prior in six cases.

   The frequency distributions of $\Delta\theta$ for different trajectory lengths (Fig. 7b) resembles the distributions obtained for the Ural events, with a bimodal structure and increased heating and cooling for longer trajectories. However, the fraction of heated trajectories (inset panel in Fig. 7b) increases more rapidly than in the Ural case, reaching 41 % already at 3 days for Scandinavian

events compared to 32 % for the Ural case. This rapid increase saturates after 5 days; by 6 days, the heating fraction of 58 % (18595 heated trajectories) is comparable to Ural case. The case-to-case variability of the computed heating fractions at 6 days is much smaller for the Scandinavian than for the Ural events: 10 of 11 Scandinavian blocks obtain fractions over 40 %, with a minimum fraction of 36 % (see Table S3 in the supplemental material, Sect. S1). As a result, the average over the per-event fractions at 6 days is actually 60 %, somewhat higher than the respective value for the Ural events. Thus, diabatic heating plays

a major role also for Scandinavian blocking events.

   At 6 days before arrival in the blocking region, about 75 % of trajectories in the heating regime are in the lower troposphere, at pressures >500 hPa (Fig. 7d). As for the Ural cases, the median trajectory is close to 700 hPa but with a higher $\theta$ of 308 K. From here, the air parcels warm and rise into the upper troposphere, obtaining a median heating of 10 K for six-day trajectories (Fig. 8b). The ascent rate within two days and the change in specific humidity along the heated trajectories are slightly smaller

in magnitude compared to the Ural ones, obtaining median values of around 300 hPa (Fig. 8f) and a drying of around 3 g kg$^{-1}$ (Fig. 8j), respectively. On the other hand, trajectories in the non-heating regime are almost totally in the upper troposphere on day $-6$ (Fig. 7f), obtaining similar cooling rates as for the Ural ones (Fig. 8b) and an insignificant change in the specific humidity (Fig. 8j).





Turning to the question of where the diabatic heating of air parcels feeding into Scandinavian blocks takes place, the spatial
distribution of peak heating location (Fig. 10b) shows that it is again mostly in the Atlantic sector, but with marked displacement
toward the south-west, in closer correspondence with the climatological distribution (Steinfeld and Pfahl, 2019). In addition,
a considerable number of trajectories undergo maximum heating over eastern North America and smaller numbers over the
eastern Mediterranean and the eastern Pacific. Nonetheless, as seen for the Ural events, the distribution shown in Fig. 10b
similarly suggests a connection between Scandinavian blocking and diabatic heating in mid-latitude Atlantic cyclones.

As for the Ural events, the maximum heating of Scandinavian blocking trajectories experiencing maximum heating in the
Atlantic sector temporally takes place in one (91 %) or two (9 %) bursts, the former being clearly larger than for the Ural
ones. Furthermore, almost half of the events in the former experience peak heating at later lags (shorter than three days,
46 %), whereas earlier lags (longer than three days) or heating around day $-3$, being favoured by the Ural events, here are
less preferred (18 and 27 %, respectively). Most of the heating takes place at later lags, closer to the blocking region, with a
median lag of about $-2$ days (Fig. 12d), which is clearly smaller compared to the median lag of $-3.5$ days obtained for the
Ural events (Fig. 12b). The bulk of the temporal distribution for peak heating defined separately for all individual trajectories
experiencing maximum heating in the Atlantic sector is shifted to slightly earlier lags, though with a median lag of $-3$ days,
as seen in Fig. 12d, right. However, Scandinavian blocking trajectories tend to experience maximum heating in the Atlantic
sector generally at later lags compared to the Ural events, which is consistent with the fact that about 40 % of the Scandinavian
blocking air parcels experience heating already in the first three days after initialization (Fig. 7b). The negative correlation we
found between the per-event time of peak heating over the Atlantic and the life-stage of the Ural blocks at the time of trajectory
initialization becomes even more pronounced for Scandinavian blocks, showing a strong negative correlation of 0.8 as seen
in Fig. 12c. Three events at the time of trajectory initialization obtain blocking sector fractions below the 90[th] percentile, but
as expected, these blocks are at their onset or early mature stage (D-index below 0.5). The individual peak lags and D-index
values for Scandinavian blocks are listed in Tables S3-S4 in the supplemental material (Sect. S1).

## 6 Linkage to mid-latitude cyclones

The previous section showed that most diabatically heated trajectories associated with both Ural and Scandinavian blocks
undergo heating and lifting in the time period between $-6$ and $-1$ days relative to blocking starting points. Figure 13 shows
the tracks of all cyclones present during this five-day interval for all Ural (a) and Scandinavian (b) events. In the North Atlantic,
the tracks are characterised by large-scale cyclonic motion around Greenland, consistent with the large-scale dipole in the SLP
anomaly composite shown in the same figure. The tracks are particularly concentrated in the region marked by the yellow box
(10 - 60° W, 50 - 70° N), positioned immediately to the north-west of the area where maximum diabatic heating is concentrated
(Fig. 10), consistent with the idea that diabatic heating occurs predominantly in the warm sectors to the south-east of the cyclone
centres. Each event has at least two (up to seven) cyclones that reside in the yellow box at some time within the relevant five-
day period and 90 % of all these events obtain at least one (up to three) cyclones located within the box at the time of peak
heating. For the remaining four events, the yellow dot is located just west or north of the box and the majority of the red-





colored cyclones either experience genesis shortly after the time of peak heating or undergo lysis just before it. Nevertheless, the majority of the yellow markers reside within or close to the yellow box, confirming the importance of cyclones for the diabatic heating experienced by the blocking trajectories.

For the 30 Ural cases, an average of five cyclone tracks per event cross the yellow box in Figure 13a during the relevant five-day window preceding blocking (involving values between a minimum of 2 and a maximum of 7 crossings in individual cases). To compare with climatology, we select 30 random winter pentads, compute the mean number of crossings per pentad, and repeat 500 times. This procedure yields a median of only four cyclone crossings per pentad (Figure 14), and shows that the 30 Ural cases constitute a rare sample with exceptionally high mean cyclone activity, well above the 99[th] percentile of the

climatological distribution. The 10 Scandinavian cases, on the other hand, show an average of only four cyclone crossings per event (interval 3 - 6); comparison with climatology using random sampling of 10 pentads shows that this average lies within the interquartile range and cannot be considered exceptional. These results indicate that serial cyclone clustering (Pinto et al., 2013) can be important for generating Arctic warm events, at least in those cases associated with Ural blocking.

Returning to Fig. 13, we note that with the exception of a few cyclones, almost all of the red tracks in the figure experience

lysis either within the yellow box or immediately to the north, north-east and west of the box; only a handful continue northward to enter the Arctic. To further quantify whether the majority of Arctic cyclones at the time of peak event undergo genesis in high-latitudes, we select cyclone tracks present during a 3-day period leading up to each of the Ural (Fig. 13c) or Scandinavian (Fig. 13d) warm events, respectively. Here, cyclone tracks observed within the polar cap ($\geq 80° \mathrm{N}$, yellow circle) during the selected time window are colored red and the genesis points of these cyclones are denoted by a red solid circle. Two Ural

and three Scandinavian events lack cyclones fulfilling the aforementioned criteria. For the remaining events, we observe that the majority of the red dots reside in the high-latitudes, where the $70° \mathrm{N}$ latitude band encompasses $87\%$ and $83\%$ and the polar cap encloses $67\%$ and $42\%$ of all genesis locations for the 54 and 12 red colored cyclones related to Ural (Fig. 13c) and Scandinavian (Fig. 13d) events, respectively. Even though Scandinavian events in general obtain less Arctic cyclones with local genesis around the peak of the event compared to the Ural events, our results presented here support the results of Messori

et al. (2018) showing that cyclones present in the high Arctic around the time of peak event are mainly locally generated within the polar cap or in the close vicinity of it. Only a few of the red-colored cyclone tracks undergo genesis in the mid-latitudes (see e.g. the two red dots at $50° \mathrm{N}$ in Fig. 13c). In fact, these two cyclones are related to event 9, of which one is shown to be responsible for the lifting of the Ural blocking trajectories in the Atlantic sector, as discussed in the case study in Sect. 3.2. Another interesting difference between these two cases is the location of the selected cyclones at the peak of each event (yellow

markers on red tracks), where $60\%$ of the 30 red cyclones existing at the peak of Ural events (Fig. 13c) reside within the polar cap, whereas already half of the four red cyclones prevailing at the peak of Scandinavian events have exited the polar cap (Fig. 13d).

As seen by the SLP anomaly composite (shading in Fig. 13c,d) representing the peak of each warm event, the region of negative SLP anomalies in the northwestern Atlantic is displaced northwards, now reaching all the way into the Arctic. On the

other hand, the region of positive SLP anomalies over the Urals in Fig. 13c becomes even more pronounced at the peak of the




warm events. For the Scandinavian events, there are two distinct centers of positive SLP anomalies; one over Scandinavia and one over Urals, the latter due to the four events from the mixed cluster.

# 7 Discussion

Previous studies show that Ural blocking enhances Arctic warming and sea-loss especially over Barents-Kara-Seas (Luo et al.,
2016b; Gong and Luo, 2017; Luo et al., 2017, 2019), and that Ural blocks are able to produce stronger sea-ice decline compared to Scandinavian blocking (Luo et al., 2019). This is in agreement with our findings, where 37 of the top-50 wintertime high-Arctic warm extreme events are attributed to blocking over the Urals, Scandinavia or over both regions simultaneously, with a majority of events featuring a strong Ural block preceding peak warming in the high Arctic. Next we discuss the sequence of dynamical processes, based on the 30 warm events associated with Ural blocking, leading to these warm extremes in the
high-Arctic (Fig. 15).

1. The first stage, "Preconditions" (Fig. 15a), comprises days 9 to 6 prior to peak Ural blocking, i.e. the time before the majority of the trajectories experience diabatic heating. This period is characterized by a subtropical high SLP anomaly and significant negative SLP anomalies over Greenland, a pattern resembling the positive phase of NAO (NAO+). The circulation pattern promotes eastward/northeastward advection of warm and moist air towards the central Atlantic. Already
at this stage, some events exhibit positive SLP anomalies over the Urals.

2. The second stage, "Heating", comprises the 5-day period (6 to 1 days preceding the peak in Ural blocking) when most trajectories experience diabatic heating (Fig. 15b). The NAO+ phase continues, with deeper negative SLP anomalies found over Greenland (Fig. 15b) and the positive SLP anomalies over north Siberia strengthen, forming a dipole over the Nordic seas conducive to advection of warm and moist air into the high Arctic.

We find that the majority (68 %) of the six-day heated Ural blocking trajectories experience lifting and maximum diabatic heating in the midlatitude North Atlantic during this 5-day period. The spatial distribution of maximum heating (Fig 10a) differs markedly from the climatological distribution presented in Steinfeld and Pfahl (2019) and regions favoured by ascending WCBs, as showed by Madonna et al. (2014). For the warm events preceded by Scandinavian blocks (Fig. 10b), on the other hand, the distribution better resembles those in the studies cited above.

Additionally, this 5-day period is characterized by anomalously high cyclone activity within a region (yellow box in Fig. 15b), consistent with the region of negative SLP anomalies, located northwest of the region favoured by maximum heating for blocking trajectories. This is consistent with the NAO+, characterized by a higher activity of intense wintertime North Atlantic cyclones (Pinto et al., 2009). The combination of NAO+ pressure pattern and Ural blocking ensures a pathway for moisture transport from North Atlantic into the Arctic and thus promotes Arctic warming and sea-ice
decline (Luo et al., 2017, 2019; Papritz and Dunn-Sigouin, 2020; Fearon et al., 2020).

3. The "Event" stage (Fig. 15c) coincides with the 1-3 day time window between the peak in Ural blocking fraction and the peak of the warm event. The lag between the peak Ural blocking and peak Arctic warming found here is in line with a





lag of four days found between Ural blocks and observed warming and sea-ice loss over the Barents-Kara Seas (Gong and Luo, 2017). The formation or strengthening of Ural blocking at this stage, subsequent to the NAO+ event noted in prior stages, is consistent with previous work showing a development of Ural blocks 4-7 days after NAO+ (Luo et al., 2016a). The combined effect of a decaying NAO+ pattern and the growing block over the Ural mountains, is to create an SLP anomaly dipole (Fig. 15c) enabling penetration of heat and moisture into the polar cap leading to Arctic warming (Fearon et al., 2020; Papritz and Dunn-Sigouin, 2020). This moisture transport is also seen in Fig. 15c, which shows significant positive moisture anomalies over the Arctic Ocean.

4. The "Post" stage (Fig. 15d), is represented by a composite over three days after the warm event. Here, the upper-level forcing is diminishing and the dipole in the SLP pattern clearly weakens, concomitant with weaker moisture anomalies over the central Arctic.

As noted by Luo et al. (2016a), the formation and strengthening of the Ural blocking anomaly subsequent to the NAO+, may be the result of wave activity propagation from the decaying NAO+ towards the Ural Mountain. Our findings do not contradict this idea, but rather provide additional insight into the amplification mechanisms of Ural blocks. Specifically, we find that diabatic heating plays a leading role in amplifying Ural blocks, with around 60 % of the air parcels in Ural blocks being subject to diabatic heating over the 6-day period prior to peak blocking. These results hint at a self-amplifying mechanism behind Ural blocks: they are potentially initiated by adiabatic wave propagation and, once established, promote further advection of low-PV diabatically-processed air into the block as it is produced in the ascending branches of North Atlantic cyclones. Further study of this self-amplifying mechanism could be an interesting avenue for future work.

Given the key role of blocking and its interaction with midlatitude cyclones as a driver for the top-50 Arctic wintertime warm extremes that we find here, another interesting question that arises is whether this is also a common feature in other, less extreme warm events in the high Arctic. Starting from a blocking perspective and applying the methods used in this study could help enhance our understanding of the processes responsible for Arctic warming. A further promising avenue for further investigation would be to study events based on anomalies in the surface energy budget rather than in temperature, as the former directly affects sea ice development and could allow for deeper insights into the dynamical processes responsible for sea ice decline in the Arctic.

## 8 Summary and Conclusions

We have investigated the dynamics behind the 50 most extreme wintertime high Arctic warm anomalies, focusing on the importance of Ural and Scandinavian blocking preceding the warm extremes. Furthermore, the dynamical processes responsible for the emergence of these events were assessed, focusing mainly on the contribution of diabatic heating in midlatitude cyclones to the formation and amplification of the blocks. We answer the questions posed in the introduction as following:

- **What is the role of blocking in driving the extreme warm events?**



Blocking plays a central role for the majority of the top-50 warm events: Composites of vertically-averaged upper-tropospheric potential vorticity anomalies over the 50 events show a prominent negative anomaly over the Ural and Scandinavian sectors. The surface expression of this pattern is a SLP anomaly dipole between a high over Siberia and a low over Greenland, promoting warm and moist air advection into the high Arctic.

- **Are there regional differences in the circulation patterns?**

We find regional differences in the circulation patterns among the different warm extremes: 30 events are associated with Ural blocking—18 of which have a block only over Urals and 12 obtaining blocking over both Scandinavia and the Urals—where the peak blocking is preceded by a NAO+ pattern. Furthermore, seven events were found to be preceded by Scandinavian blocking and two by strong blocking over the Pacific, whereas 11 events–resembling the structure of Ural events but with a weaker amplitude–were not associated with any block in the three favoured blocking regions.

- **What is the importance of diabatic processes in driving the blocks?**

Diabatic heating plays an important role in the dynamics of high-latitude blocking associated with Arctic warm events: analysis of Lagrangian back-trajectories from Ural blocks show that 59 % of the air parcels experience lifting (median ascent of 363 hPa within two days) and diabatic heating (median heating of 11 K) within six days prior to arrival at the block. Almost half of the air parcels making up the negative VAPV anomalies of Scandinavian blocks experience heating already within the 3-day journey into the blocking region, reaching up to 58 % over six days. The strongest heating is

found over a region in the North Atlantic, temporally taking place at a median day of 3.5 and 2.75 prior to arrival into the Ural and Scandinavian blocks, respectively.

Furthermore, we find that the time of peak heating within the North Atlantic region, defined per each Ural and Scandinavian event, and the life-stage of the blocks are negatively correlated, indicating that younger blocks experience maximum heating preferably at later times, closer to the blocking region and vice versa.

- **How do cyclones and blocks interact during the events?**

We find a strong interaction between midlatitude cyclones and Eurasian blocks as driver of wintertime Arctic warm extremes: an exceptionally high midlatitude cyclone activity–coinciding both spatially and temporally with the time window of maximum heating for blocking trajectories in the North Atlantic—highlights the importance of latent heat release in cloud-diabatic processes ahead of strong surface cyclones in providing low-PV air into upper-level block, thus

enhancing and amplifying the high-latitude block. On the other hand, these cyclones are also guided polewards by the block, further promoting northward transport of heat and moisture and thus helping generate the Arctic warm extremes. These midlatitude cyclones mostly decay before entering the high Arctic, whereas around the time of each warm event, a peak in locally-generated polar cyclone activity is observed. For Scandinavian events, on the other hand, the cyclone activity is not as exceptional.

This study deepens the understanding of the underlying processes driving the warming seen in the high Arctic, emphasizing the importance of atmospheric blocks and their tight interaction with midlatitude cyclones–as amplifiers of the block or being



guided by the block–as well as the combined effect of the prevailing circulation patterns on the appearance of high Arctic extreme warm events. It also highlights processes that need to be well captured in models to be able to represent the Arctic wintertime climate.

*Code availability.* Code for the analysis is available upon request.

*Data availability.* ERA-Interim data can be obtained from ECMWF via https://www.ecmwf.int/en/forecasts/datasets/archive-datasets/reanalysis-datasets/era-interim.

*Author contributions.* SM, GS, and RC developed research ideas and designed the study. LP helped with the trajectory calculations. The analysis was performed by SM while continuously discussing results with RC and GS. SM wrote the initial draft of the manuscript and all
authors contributed with feedbacks and reviews to the sections. The final draft was edited by all authors, providing with suggestions and improvements for the final version.

*Competing interests.* The authors declare that they have no conflict of interest.

*Acknowledgements.* This work was funded by Knut och Alice Wallenbergs Stiftelse, Grant/Award Number: 2016-0024. We thank Erica Madonna (University of Bergen) for providing us with the cyclone tracks and ETH Zürich, especially Michael Sprenger, for all the technical
support and help in the setup for the LAGRANTO Tool as well as for their inspiring and helpful advice. We are also grateful for helpful discussions with Timo Vihma and Tiina Nygård (FMI). The data analysis and visualization in this study was done using Python.



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



**Figure 3.** Synoptic situation prior to three warm extreme events (columns) between 3 and 24 January 2006 at 1200 UTC at lags $-5$, $-3$, $-1$ and 0 days relative to each warm event (rows). Panels a-l show potential temperature ($\theta$) on the dynamical tropopause, defined by the 2-pvu isosurface (shading), SLP (hPa, black contours every 10 hPa, solid for SLP > 1000 hPa), total column water (thick black solid for 5 kg m$^{-2}$) and T2m (purple solid at 0° C). Dotted areas indicate regions with identified blocks. Magenta and red boxes indicate the Ural (h,f) and Scandinavian (g) blocking sectors, respectively. Panels m-o show T2m anomalies (shading) and SLP contours at 1200 UTC of each warm event (lag 0 day). The black circle shows the latitude line at 80° N.



**Figure 4.** Six day back-trajectories initialized from the blocks (black hatched regions, first row) prior to the three warm events (columns). Rows indicate the trajectory locations (pink dots) at the starting point (a-c), the time of maximum diabatic heating (d-f), and the end point (g-i). The dates for which the panels are valid are denoted by lag days relative to the peak of the warmest event 1. Furthermore, the trajectories are drawn between the valid time and the starting time, colored according to pressure. Additionally shown are total column water (gray shading, $\mathrm{kg\,m^{-2}}$), SLP anomalies (hPa, black contours ever 5 hPa, dashed for < 0 hPa, zero contour not shown) and cyclone location at valid time (green dot) and tracks (green lines). Further shown is the density of trajectories experiencing maximum heating at the valid time (green contour for 5 % $(10^6\,\mathrm{km^2})^{-1}$). The relative percentage of all heated trajectories experiencing their maximum heating at the valid time is given in the text boxes (second and third rows). The latitude circle at 40° N is indicated by a black dashed line and the magenta and blue boxes indicate the Ural and Scandinavian blocking sectors, respectively.

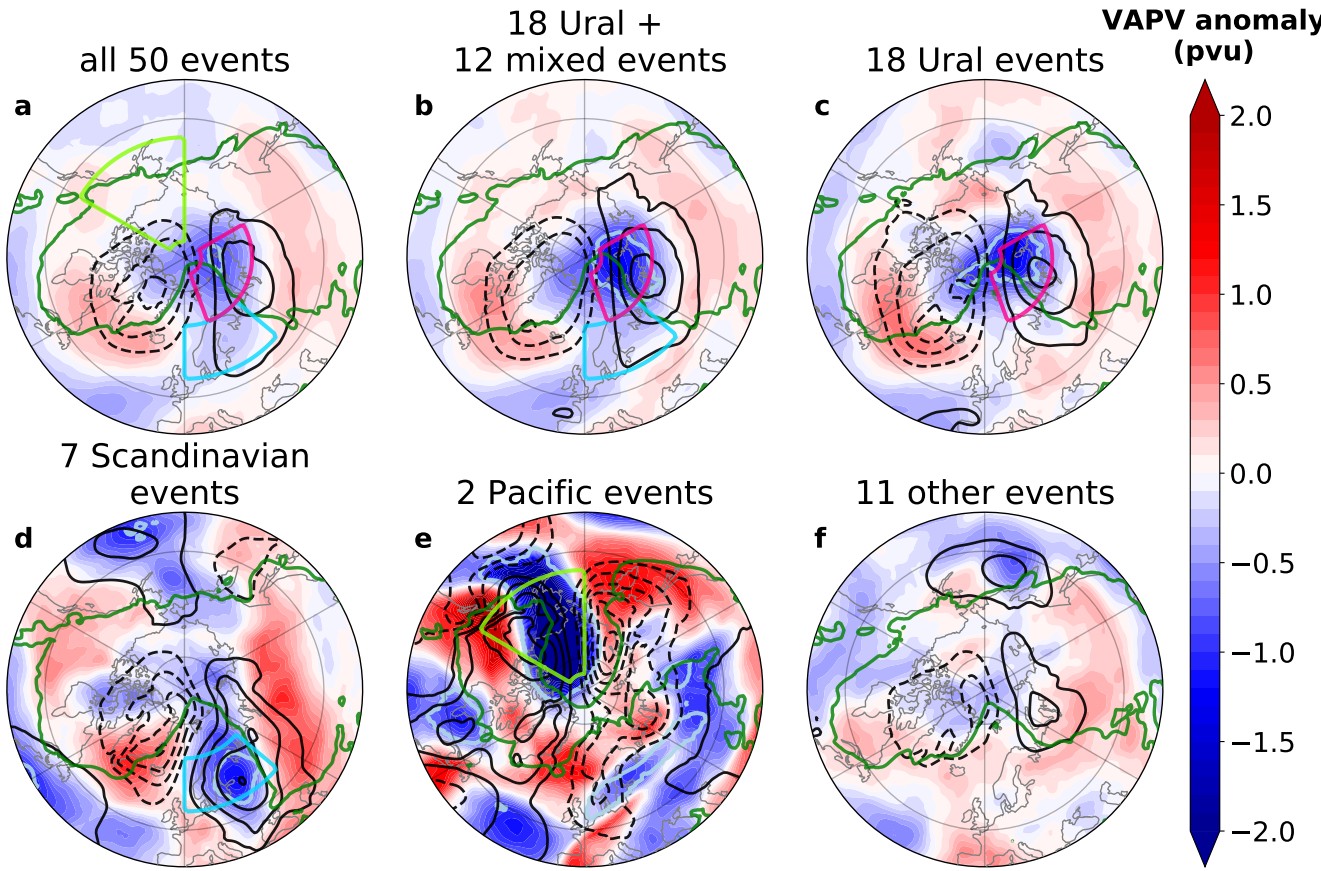

**Figure 5.** Circulation patterns observed during warm extremes shown as time mean composites over days $-3$ to $-1$ prior to the warm events for (a) all 50 extreme events, (b) 30 events with Ural blocking, comprised of 18 pure Ural events and 12 events within the mixed cluster, (c) 18 pure Ural events, (d) 7 pure Scandinavian events, (e) 2 Pacific events, and (f) remaining 11 events belonging to the other cluster. Shown are VAPV anomalies (pvu, shading) overlaid with SLP anomalies (hPa, black contours every 5 hPa, dashed for $< 0$ hPa, zero line not shown), total column water (green solid isoline for 5 kg m$^{-2}$) and blocking frequency for the events and respective time steps involved in each composite (light blue solid contour for 33 %). The colored boxes in (a) show the Ural (magenta), Scandinavian (blue) and Pacific (light green) sectors.



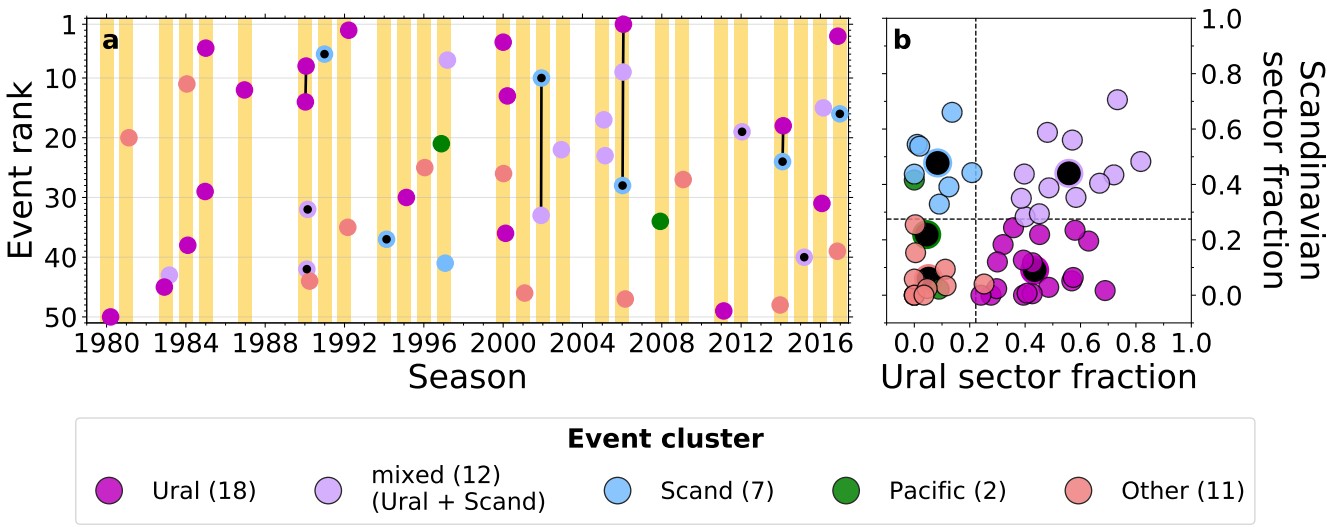

**Figure 6.** Classification of the top 50 warm extremes by type of blocking. (a) The 50 events ordered by time and ranked by the strength of the temperature anomaly. Events discussed in the Scandinavian event analysis (Sect. 5.2) are marked with a black dot. Multiple warm events associated with a single blocking event are connected by a thick black line. Yellow bars mark the NDJFM season for each year (where e.g., 1980 refer to winter $-79$ to $-80$). (b) Scatter plot of Ural vs. Scandinavian daily blocking index for the 50 warm events. Smaller circles mark the maximum index value in the 6 days preceding each event; large circles with black center indicate mean values for each cluster. Dashed lines show $90^{\text{th}}$ percentile threshold for each index. The marker colors refer to the five different event clusters (see legend).

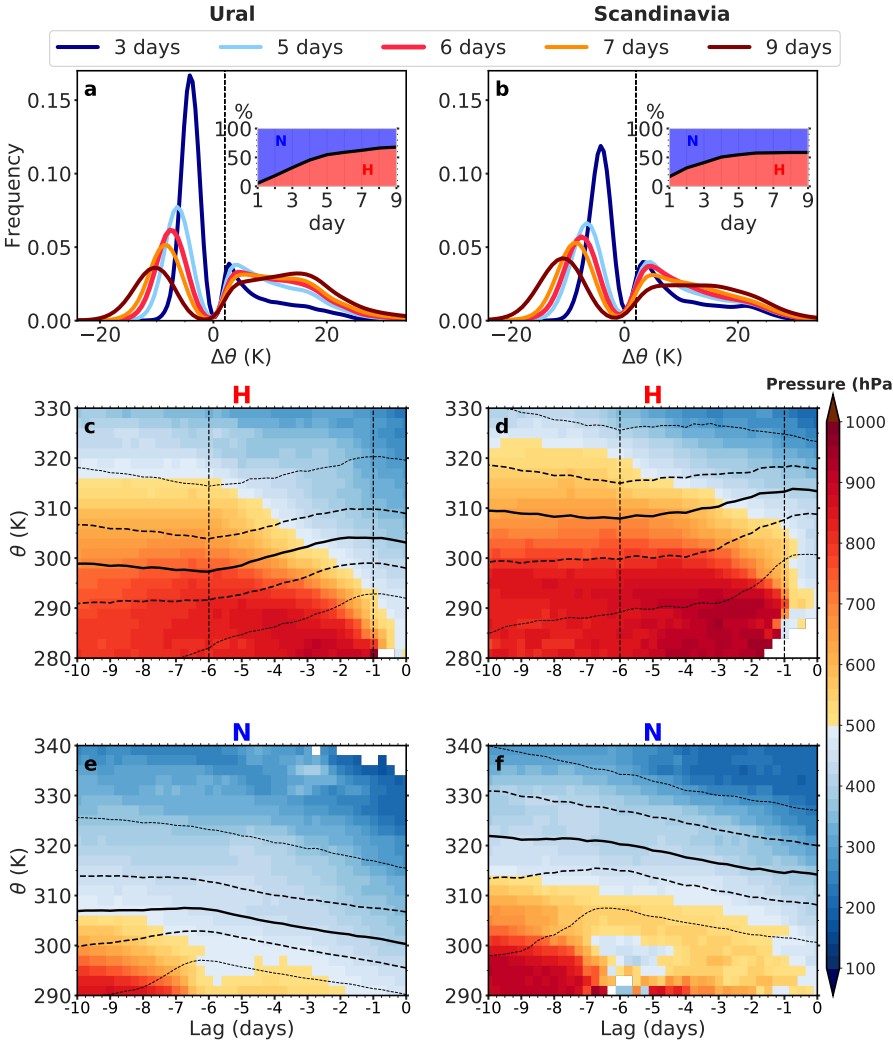

**Figure 7.** Ural (first column) and Scandinavian (second column) blocking trajectory regimes and their characteristics. (a,b) Frequency distribution of maximum change in potential temperature along back-trajectories initialized from Ural (a) and Scandinavian (b) blocks during three (blue), five (light blue), six (pink), seven (orange) and nine (brown) days before their arrival into the blocking region. The threshold value of $\Delta\theta = 2\,\mathrm{K}$ is shown by the black vertical dashed line. The inset figure shows the percentages of trajectories counted in the no-heating (N, blue) and heating (H, red) regimes by increasing length of trajectories. Figure is inspired by Steinfeld and Pfahl (2019). (c-f) Pressure evolution (hPa, shading) with potential temperature on the vertical axis of blocking trajectories within the two regimes: heating (H, in red) (c-d), and no-heating (N, in blue) (e-f), averaged within each grid box ($6\mathrm{h} \times 2\,\mathrm{K}$). Note the different y-axis range for the two regimes. The distribution of trajectories is shown in black lines (median in solid, thick dashed lines for the IQR and the thin dashed lines for the $5^{\mathrm{th}} - 95^{\mathrm{th}}$ percentile range). The regime separation in (c-f) is based on six-day back-trajectories. The two vertical black dashed lines in (c-d) mark the lifting window between lags $-1$ and $-6$ days.

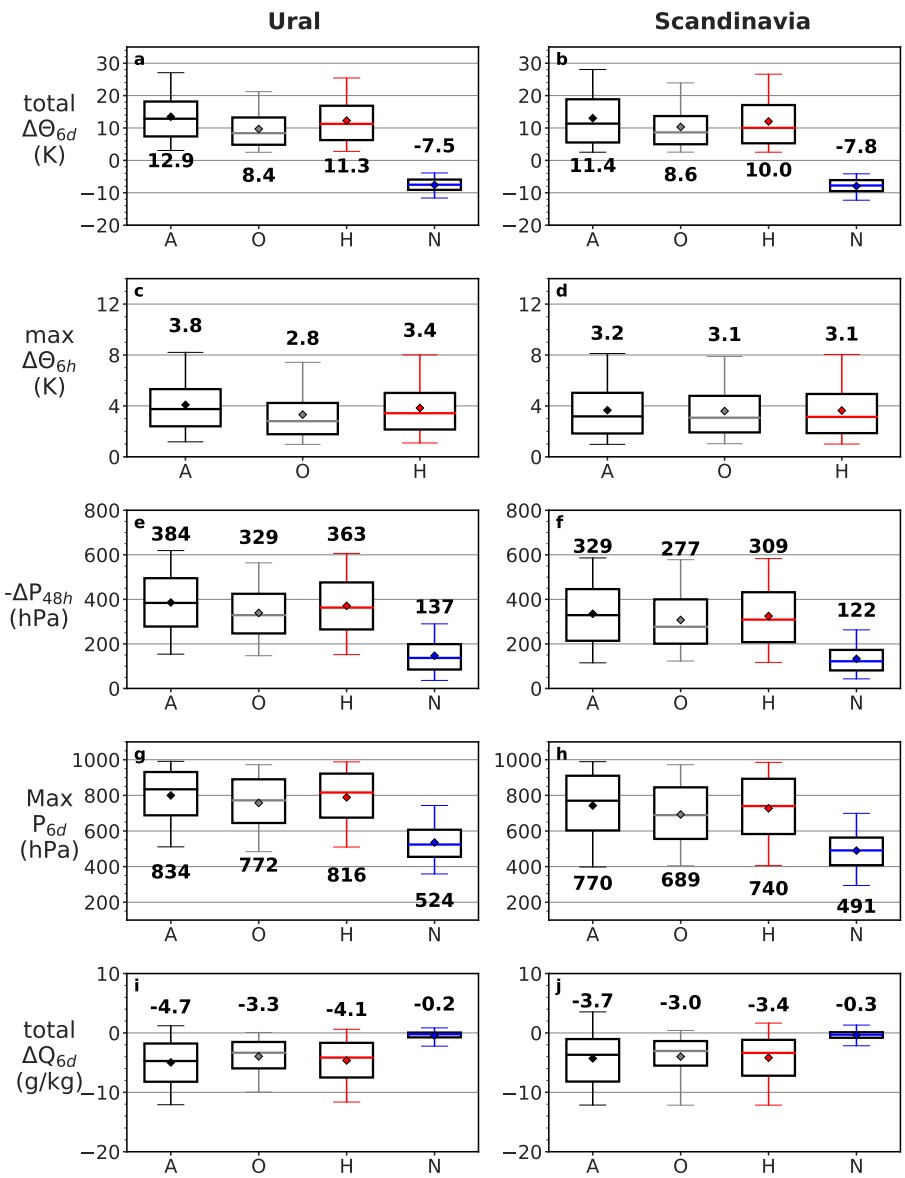

**Figure 8.** Statistical distributions of maximum absolute change in potential temperature (K) (a, b), maximum 6-hourly change in potential temperature (K) (c, d), maximum absolute ascent (hPa) within 48-hours (e, f), maximum pressure (hPa) (g,h) and maximum change in specific humidity ($g\,kg^{-1}$) (i, j) along the 6-day back-trajectories belonging to the "non-heating" regime (N, blue lines) and "heating" regime (H, red lines), initialized from both Ural (first column) and Scandinavian (second column) blocks. The heating regime is further sub-divided into trajectories experiencing maximum heating in the main domain (A, black lines) and outside the domain (O, gray lines). The second row is only shown for heated trajectories. The whiskers show the $5^{th} - 95^{th}$ percentile range, the box the interquartile range and the mean is shown as diamonds. Horizontal lines as well as the bold values denote the median for each distribution.



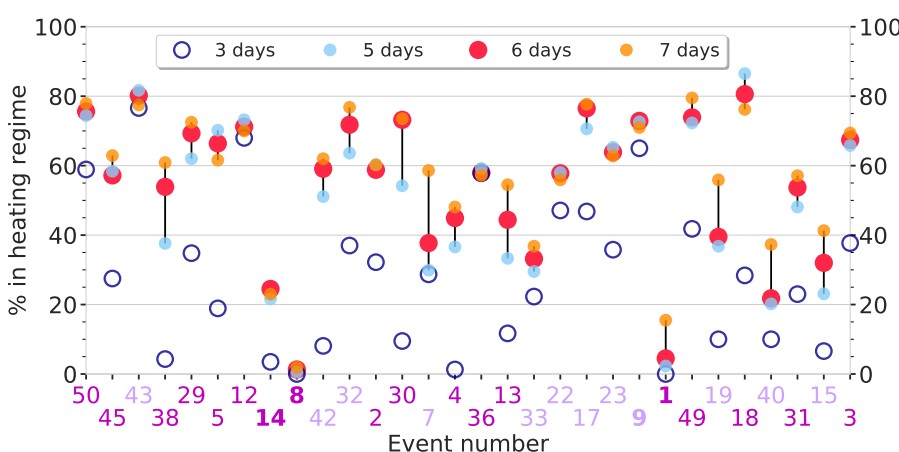

**Figure 9.** Percentages of blocking air parcels in the heating regime shown per Ural event for back-trajectories during three (blue), five (light blue), six (pink) and seven (orange) days before their arrival into the blocking region. The percentage for five and seven days are connected with a black vertical line. Event numbers refer to event ranking, here sorted by time, where magenta numbers refer to pure Ural events (Ural cluster), light purple to the mixed cluster and bold numbers (Event 14 and 8, Event 9 and 1) refer to events associated with the same block.

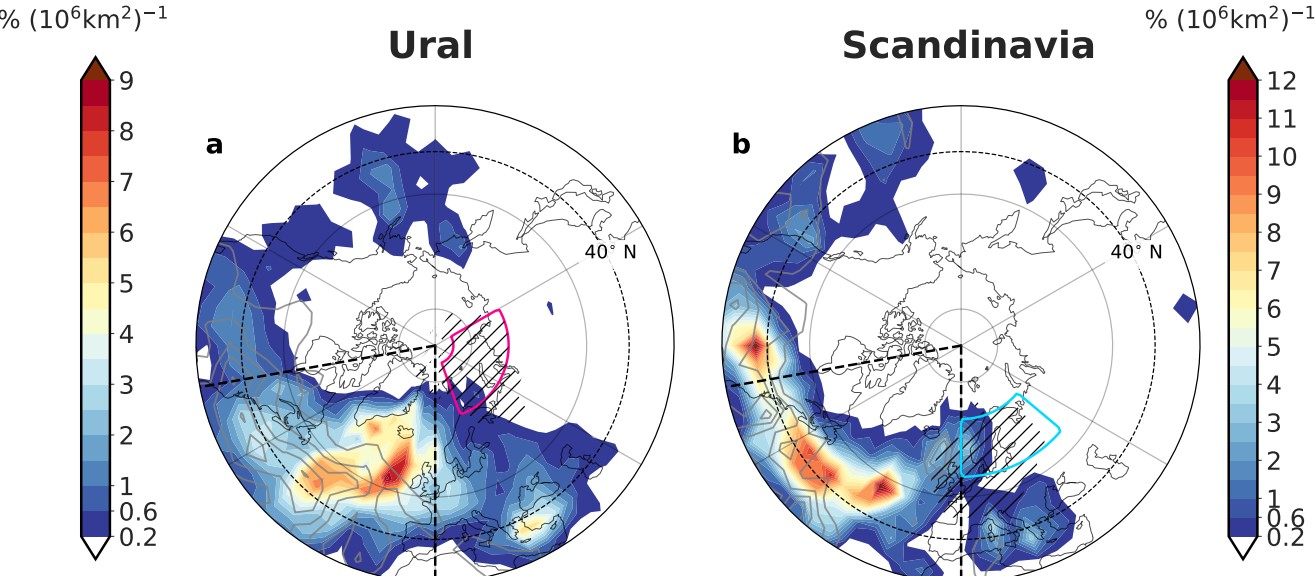

**Figure 10.** Spatial density distribution of the maximum heating for the trajectories within the heating regime initialized from Ural (a) and Scandinavian (b) blocks. Shading shows the density distribution for the location of the maximum 6-hourly heating, normalized by the total number of heated trajectories and area weighted by the grid cell area ($5° \times 5°$). The gray contours show the locations of the trajectories six days prior to arrival into the blocking region (normalized area-weighted density, shown every 1, starting from $1\,\% \,(10^6 \,\mathrm{km}^2)^{-1}$). The magenta box in (a) and the cyan box in (b) show the Ural and Scandinavian sector, respectively, where the hatched regions indicate the main backward trajectory starting locations within the blocks (normalized area-weighted density $> 5\,\% \,(10^6 \,\mathrm{km}^2)^{-1}$). The main heating domain is denoted by the black dashed lines and the black dashed circle shows the latitude line at $40°$ N.



Weather and Climate Dynamics Discussions — Open Access — EGU

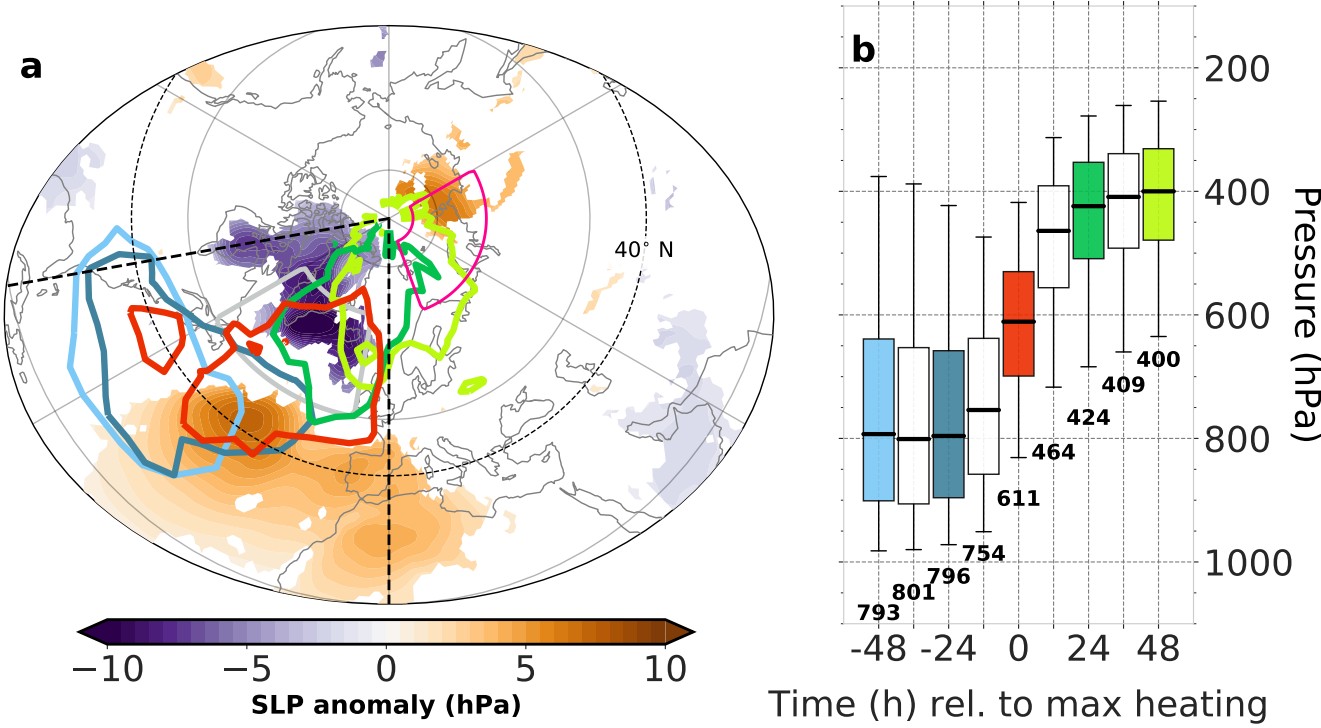

**Figure 11.** Spatial and vertical evolution of Ural blocking trajectories belonging to the main heating domain (region denoted by the black dashed lines in (a)). (a) Significant SLP anomalies two days prior to the timing of by event defined peak frequency in max heating within the main heating domain (hPa, shading, significant when $\geq 67\%$ of the 30 members obtain the same sign in the anomaly as the composite mean), overlaid with density contours of trajectories at the location of max heating (red), one day before (turquoise), two days before (light blue), one day after (green) and two days after (light green), normalized by the total number of trajectories included at the corresponding time step and area-weighted by each grid cell area $(5° \times 5°)$. Each density contour shows the $4\% \, (10^6 \, \text{km}^2)^{-1}$ value. The magenta box denotes the Ural sector, i.e., the origin region for the blocking trajectories, and the gray box the region used for analyzing cyclone activities. The dashed black circle shows the latitude band at $40°$ N. (b) Evolution of pressure along trajectories relative to the time of maximum heating within the main A-domain, coloring as in (a). Median values per 12-hourly time slots are shown in text and as horizontal lines, boxes denote the interquartile range and whiskers show the $5^{\text{th}} - 95^{\text{th}}$ percentile range.



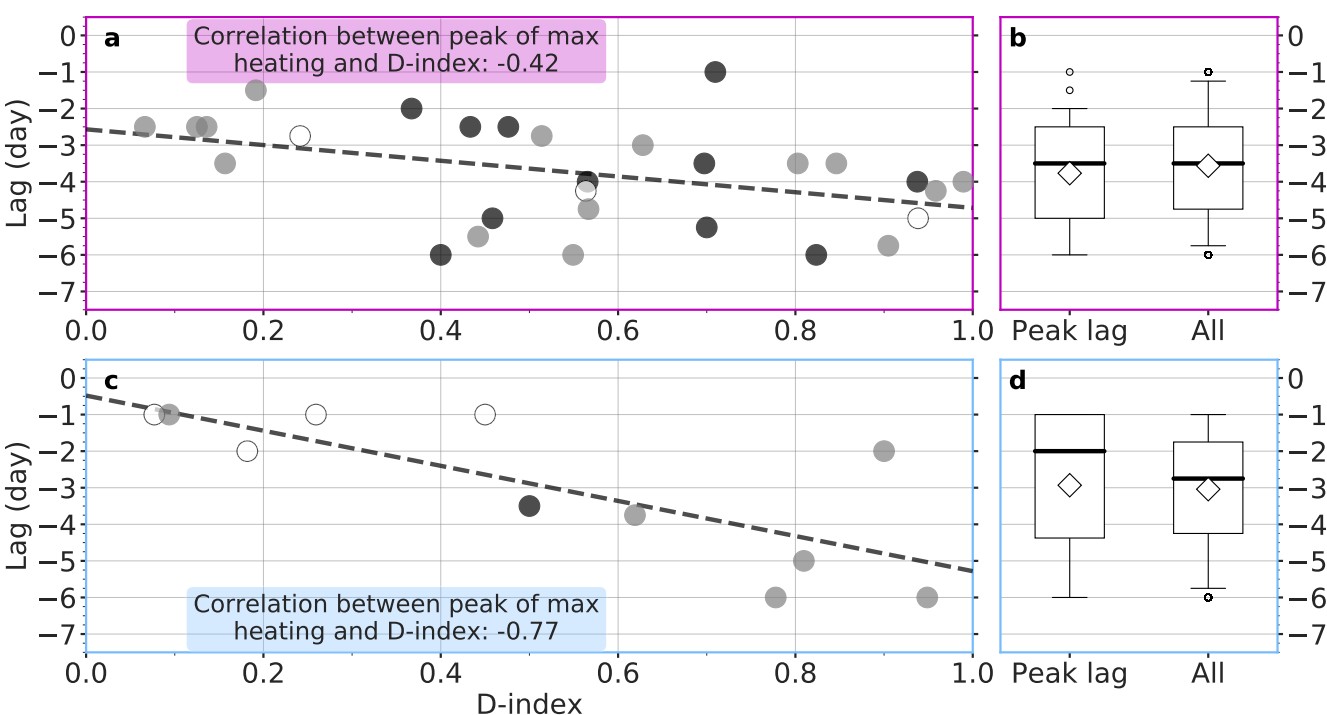

**Figure 12.** Correlation between life stage of the Ural (first row, magenta) or Scandinavian (second row, light blue) blocks and the time lag of peak heating within the main Atlantic (A)-heating domain (a, c), markers colored by the Ural (a) and Scandinavian (b) sector blocking fraction (dark gray marker > 99[th] percentile, white marker < 90[th] percentile and gray in between) at trajectory starting points (see also Tables S1-S4 in the supplemental material, text section S1). The life stage of the block at the time of trajectory initialization is given by the D-index (see Eq. (1)). (b, d) The distribution for the time of peak heating in the Atlantic sector (total of 29 blocks in (b) and 11 blocks in (d)) and the distribution of the time lag of peak heating defined separately for all trajectories experiencing maximum heating in the selected Atlantic sector (total of 38825 trajectories in (b) and 11692 in (d)) are shown in boxplots. The horizontal line denotes the median, the diamond the mean, the box the interquartile range, the whiskers the 5[th] - 95[th] range and the circles the outliers.





**Figure 13.** Cyclone track distribution around warm extremes related to Ural (left column) or Scandinavian (right column) blocks. (a,b) Cyclones observed within lags $-1$ to $-6$ days relative to trajectory initialization (blocking region), shown for 30 Ural (a) and 10 Scandinavian (b) events, respectively. Tracks are colored red if the selected cyclone is both in temporal and spatial correspondence with the considered time period in the yellow sector (C, $10 - 60°$ W, $50 - 70°$ N). If there is only a temporal match, the cyclones are colored gray. Shading shows the SLP anomaly composite over these events at the case-defined peak of maximum heating within the main heating domain for blocking trajectories (for 29 Ural and 10 Scandinavian events in (a) and (b), respectively, see also Tables S2 and S4 in the supplemental material (Sect. S1). Yellow circles show the location of the red-colored cyclones at this peak of maximum heating. Red or gray solid triangles show the lysis of cyclones that cross or stay outside the chosen sector, respectively. (c, d) Same as in (a, b) but shown for cyclones observed up to three days prior to the peak of each Ural (c) or Scandinavian (d) event, respectively, where the red coloring refers to cyclones that reside in the high Arctic ($\geq 80°$ N, yellow latitude band at $80°$ N) within the time period considered. The SLP anomaly composite field and yellow circles for the red-colored cyclones are shown at the time of each warm event. Red and gray solid circles denote the genesis of cyclones that cross or stay outside the chosen sector, respectively.



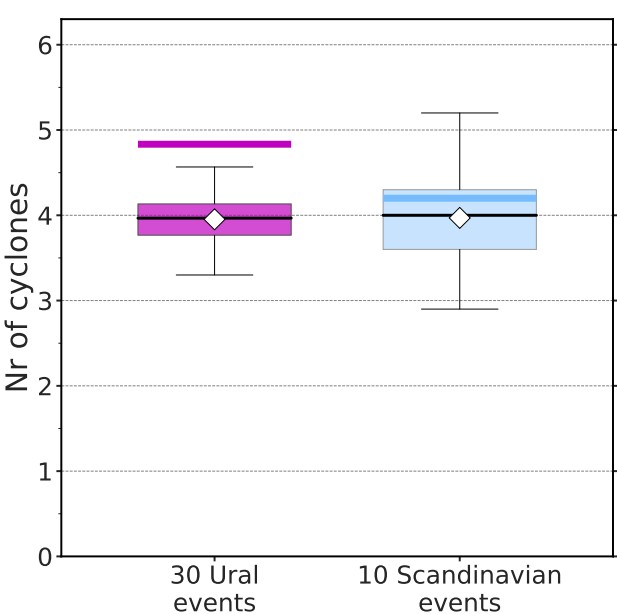

**Figure 14.** Cyclone climatology presented as the average cyclone frequency observed in the chosen sector (C, yellow sector in Fig. 13a,b, 60-10° W, 50-70° N) northwest of the main heating region during a 5-day window (lags $-1$ to $-6$ day relative to trajectory initialization within the block) for 30 Ural (magenta) and 10 Scandinavian (light blue) events, computed using Monte Carlo re-sampling of 30 or 10 arbitrary "events" within the analysis period with 500 repetitions, respectively. The black horizontal line denotes the median, mean is shown with diamonds, the box refers to the IQR and the whiskers the $1^{st} - 99^{th}$ percentile range of the distribution. The magenta and light blue horizontal line at each boxplot denote the cyclone frequency for the 5-day window averaged over the 30 or 10 chosen Ural and Scandinavian events, respectively.



**Figure 15.** Chain of processes related to Ural warm events as time and event composite during four identified time periods: at preconditions (days −9 to −6 relative to trajectory initialization, 3 days per event) (a), at heating (−6 to −1 days with respect to trajectory initialization, 5 days per event) (b), in the close vicinity of the warm events (days between lag −1 day relative to trajectory initialization until the peak of warm event, 2-4 days per event) (c) and at post conditions (up to three days after each warm event, 3 days per event) (d). Significant total column water anomalies shown in shading (kg m$^{-2}$, significant when > 67 % of the 30 members obtain the same sign in the anomaly as the composite mean), overlaid with SLP anomalies (hPa, every 5 hPa, significant in bold contours, red for positive, blue for negative anomalies, zero anomaly contour is not shown) and potential temperature on the 2 pvu surface (purple solid isoline for 310 K). The black dashed region in (b) denotes the main heating region, the yellow sector in (b) the cyclone region and the dashed black circle shows the latitude band at 40° N.