# Peer review of "Interaction between Atlantic cyclones and Eurasian atmospheric blocking drives wintertime warm extremes in the high Arctic"

_Weather and Climate Dynamics, 2021_

## Author Response (AR1)

Dear Reviewer,

We appreciate and thank you for your comments. Please find below some thoughts, replies and specifications of changes we made as a response to your suggestions. All changes and added sentences based on suggestions imposed by both RC1 (you) and RC2 are highlighted in red in the revised manuscript.

**Major comments:**

**(1) As suggested by the thermal budget analyses in Kim et al. 2021, the arctic temperature changes are primarily driven by the temperature advection associated with the SLP anomalies over the Ural region and partly cancelled by the diabatic heating. Wang et al., 2021 have suggested that diabatic heating acts as a damping mechanism for the Ural blocking both upstream and downstream. It seems important to make a comparison of the role of diabatic heating in this work to those results.**

Thank you for indicating this to us and for referring us to inspect the recent studies, one focusing on the drivers behind Arctic (and Eurasian) temperature changes and their linkages to anticyclonic anomalies over the Ural region and the other pointing out the various processes acting on the blocking evolution. We have addressed this as following: added a few sentences in Sect. 1.1 to clarify the links between circulation anomalies and their concomitant behaviour, especially highlighting the importance of Ural blocking or exeptional anticyclonic anomalies on the sea ice melt and warming in the BKSs by enhancing moisture flux convergence or inducing northward temperature advection (L68). The influence of blocks on the surface warming is further clarified. This issue is further highlighted in the Section 7 (L527, L560).

Regarding the comparison of diabatic heating implying either an amplifying or dampening effect on the blocking dynamics, as well as the discussion of other mechanisms related to cyclones as responsible for the development of blocks, we added and modified as following:
Section 1.2 highlights the diverse processes responsible in the blocking evolution (L87), further linking the role of diabatic heating on the blocking dynamics to the stage of the blocking life-cycle in order to place the context of our results to recent studies (L105). We return to the comparison of our findings and the resent studies shortly in (L369) and further more comprehensively in the Section 7 with a paragraph starting at (L580), preceded by a discussing the link between adiabatic and diabatic processes responsible for blocking dynamics in (L571).

**(2) The authors highlight the role of diabatic heating associated with the North Atlantic cyclones in affecting the Ural and Scandinavia blockings, however, the mechanisms of which are not comprehensively discussed in the current manuscript. They focus on the sources of low-PV air associated with the blockings and show that ascending warm air associated with Atlantic cyclones may influence the high-latitude blocking. I believe adding more discussions on the dynamics of heating from North Atlantic affecting the Ural blocking are necessary for publication in "weather and climate dynamics".**

We appreciate your opinions and have added more discussion about the mechanisms related to cyclones and their interaction between blocks, as given more in detail in the reply for comment (1). We have additionally cited a study, where they by numerical experiments show the impact of turning off the latent heating upstream blocking on blocking development (L105, L572).

**(3) The title does not fully reflect the essence of the article. In the current manuscript, the authors highlight that diabatic heating associated with the Atlantic cyclones can affect the Eurasian**

**atmospheric blocking. They didn't show the impact of Eurasian blocking on the Atlantic cyclones. There are no discussions on the interaction between Atlantic cyclones and Eurasian atmospheric blocking. In this regard, the authors are encouraged to correct the title, e.g., to adjust it, excluding the mention of the interaction.**

We appreciate for emphasising this specific point to us. However, we do show the interaction both ways: cyclones help amplify and maintain blocks as injecting low-PV air to the upper troposphere via diabatic heating, as you nicely point out in your comment, but blocks are also guiding the cyclones northwards, thus bringing warm and moist air into the Arctic and help generating the warm events. We have clarified this connection further in the conclusions (L630) and slightly prior at (L588). As suggested by RC2, we decided not to reformulate the title but add "wintertime" before warm extremes to emphasize their occurrence during winter seasons.

**(4) In section 3, the authors show the trajectory classification results of three warm events in January 2006. However, the details of the methodology are introduced in section 5. I thus recommend the authors to move the trajectory classification methodology to section 2. It is more fluent for the manuscript flow and easier for the reader to understand the results in section 3.**

This issue has been discussed a few times between the authors of the paper while writing the manuscript and in the beginning, we also had the classification of trajectories introduced in Section 2. However, we came to the conclusion to keep it in Section 5 where it is more in its context and added a small explanation in Section 3 (L236) following your remark. We appreciate your aspect regarding this.

**Minor comments:**

**Figure 1: the dashed cyan lines are hard to recognize as the background shading is already blue.**

We have changed the cyan dashed lines to orange coloured.

**Figure 3 has too much information in panels (a)-(l). The green and purple lines as well as the magenta and blue boxes are hard to recognize in those plots. I thus recommend the authors to replot these panels.**

We received a similar comment before final submission of the manuscript and modified the contours as following: green line (total column water) changed to black and blue boxes (Scandinavia) to red. We kindly ask you to check the newest version obtaining these changes.

**Line 340/420, since many other processes (e.g., horizontal advection of temperature in Kim et al., 2021, barotropic and baroclinic processes associated with local wave activities in Wang et al., 2021) are not discussed in this section, the conclusion of "diabatic heating plays a major role in the blocking events" should be cautious.**

As suggested by RC2, we have reformulated the phrases stating that "diabatic heating plays a major role in the blocking events" in order to more devotedly represent the results we retain (as in L363, L443 and L614).

Kind regards,
Sonja Murto & co-authors

Dear Reviewer,

We appreciate and thank you for your comments. Please find below some thoughts, replies and specifications of changes we made as a response to your suggestions. All changes and added sentences based on suggestions imposed by both RC1 and RC2 (you) are highlighted in red in the revised manuscript.

**The manuscript is clear, and the results are interesting. The manuscript builds on previous work looking at the role of diabatic heating on blocking more generally, with similar methods, but extends it by focussing on the important impact of Arctic warming. As noted by another reviewer, there is a body of literature specifically on the dynamics of Ural blocks and their impact on the Arctic, and the new results here should be placed into the context of these studies in more detail (specifically, how diabatic heating and cyclone interaction relate to other physical mechanisms of block development); however, since other comments already raise this issue, I do not pursue it further. With this caveat, I recommend the manuscript be accepted for publication once the following comments have been addressed.**

We appreciate your remark on this issue, as also raised by other comments, and have added clarifying and complementing paragraphs and sentences in Sections 1.1, 1.2 and 7, further discussing the various processes involved in Ural blocking development (L87, L568) and the importance of diabatic heating as amplifier of the blocks (L105, L572) or its decay (L580). Furthermore, the impact of blocks on the temperature anomalies in the Arctic is further enriched with more recent literature, as is found in the introduction (L68) and in the discussion (L527, L560) sections.

**General comments:**

**Title: The study focusses on wintertime warm extremes, and this should be mentioned in the title.**

We have followed your advice and added "wintertime" prior to "warm extremes" in the title.

**L5: The period of study should be mentioned here (1979-2017).**

We agree on this and have added "within the period 1979-2016." In the end of the sentence in (L5).

**L13: 'the contribution of diabatic heating to these blocks is around 60%' does not make sense. Please be more precise.**

We have changed "; the contribution of diabatic heating to these blocks is again around 60 % for six-day back-trajectories," to " Around 60 % of the six-day back-trajectories started from these blocks experience diabatic heating,".

**L89 and L95: 'wintertime Arctic warm extremes' is more accurate than 'Arctic warm extremes' (unless there are no warm extremes outside of winter, according to your definition, in which case this would be worth mentioning).**

Thank you, we have added the word "wintertime" (now L110 and L117).

**Fig 1 caption: It took me a while to see the horizontal blue line (the cyan and light blue lines look very similar, especially against the blue shading!). Is there a better choice of colours?**

Based on the comments from R1, we have changed the cyan dotted line to orange dotted line (see new figure in the revised manuscript).

**L135: For clarity, is the overlap condition based on number of grid points or area?**

The overlap condition is based on the blocking mask, which consists of grid points of zeros and ones.

**L152: Please describe the release grid more precisely.**

We added a clarification regarding the release grid of trajectories: "More specifically, trajectories are initialized at every grid point within the blocking mask, horizontally equally at every 80km × 80km grid and vertically every 50hPa between 500 and 150hPa" (L173).

**Fig 10 caption: I found the terminology here confusing because word the density is ambiguous. Consider rewriting (e.g. perhaps something like: 'Spatial distribution of the locations of maximum heating for the trajectories within the heating regime initialized from Ural (a) and Scandinavian (b) blocks. Shading shows the density of trajectories at the time of maximum 6-hourly heating, defined as the percentage of the total number of heated trajectories per unit area. …').**

We agree with you and have changed the figure caption to: "Spatial distribution of the locations of maximum heating for the trajectories within the heating regime initialized from Ural (a) and Scandinavian (b) blocks. Shading shows the density of trajectories at the time of maximum 6-hourly heating, defined as the percentageof the total number of heated trajectories per unit area (here5°×5°)."

**Line 499: 'sea-loss' -> 'sea-ice loss'**

Thank you for noticing our mis-spelling. We have changed it accordingly (L523).

**L564: I do not think your results prove that 'Diabatic heating plays an important role in the dynamics of high-latitude blocking', as claimed. You have shown very clearly that most air parcels entering the blocks do undergo diabatic heating, but of course that does not necessarily mean that the diabatic heating is important for the block evolution. Indeed, you have selected cases whereby warm moist air moves north and enters the Arctic, and it is hard to envisage that happening without diabatic heating occurring. Having said that, I do agree it is certainly likely to play a role. But the language used here should represent the results of the paper more faithfully.**

We agree and have rephrased the sentences in the paper where we stated "diabatic heating plays an important role in the dynamics of high-latitude blocking" (now L614, as well as L363 and L443). Furthermore, based on your comment, we have added new sentences in the introduction and discussion sections to clarify the role of diabatic heating associated with Ural blocks (see red text).

Kind regards,
Sonja Murto & co-authors

---

## Author Response (AR2)

Dear reviewers and co-Editor,

We appreciate for your helpful comments. Following the suggestions imposed by Referee 1 and the co-Editor, we have slightly modified and added one sentence to the abstract clarifying the interaction we are highlighting in our paper (L1, L16-19).

Furthermore, we have modified Figures following the suggestions from the co-Editor:
- Figure 3 – we changed the black-dotted areas showing the blocking regions to white dots and further encircled by a white contour.
- Figure 15b – we changed the brightness and line thickness the yellow box to be more visible.

For the changes, please see the red-highlighted text of the revised manuscript.

Kind regards,

Sonja Murto & co-authors